# Zebrafish as a Versatile Model for Cardiovascular Research: Peering into the Heart of the Matter

**DOI:** 10.3390/cells14070531

**Published:** 2025-04-02

**Authors:** Ramcharan Singh Angom, Meghna Singh, Huzaifa Muhammad, Sai Manasa Varanasi, Debabrata Mukhopadhyay

**Affiliations:** 1Department of Biochemistry and Molecular Biology, Mayo Clinic, College of Medicine and Science, Jacksonville, FL 32224, USA; angom.ramcharan@mayo.edu (R.S.A.); hmuhammad@alfaisal.edu (H.M.); varanasi.saimanasa@mayo.edu (S.M.V.); 2Department of Pathology and Lab Medicine, University of California, Los Angeles, CA 92093, USA; meghnasingh@g.ucla.edu; 3College of Medicine, Alfaisal University, Riyadh 11533, Saudi Arabia

**Keywords:** zebrafish, cardiovascular disease, CRISPR, TALEN, heart development, mutant, forward genetics, reverse genetics

## Abstract

Cardiovascular diseases (CVDs) are the leading cause of death in the world. A total of 17.5 million people died of CVDs in the year 2012, accounting for 31% of all deaths globally. Vertebrate animal models have been used to understand cardiac disease biology, as the cellular, molecular, and physiological aspects of human CVDs can be replicated closely in these organisms. Zebrafish is a popular model organism offering an arsenal of genetic tools that allow the rapid in vivo analysis of vertebrate gene function and disease conditions. It has a short breeding cycle, high fecundity, optically transparent embryos, rapid internal organ development, and easy maintenance. This review aims to give readers an overview of zebrafish cardiac biology and a detailed account of heart development in zebrafish and its comparison with humans and the conserved genetic circuitry. We also discuss the contributions made in CVD research using the zebrafish model. The first part of this review focuses on detailed information on the morphogenetic and differentiation processes in early cardiac development. The overlap and divergence of the human heart’s genetic circuitry, structure, and physiology are emphasized wherever applicable. In the second part of the review, we overview the molecular tools and techniques available to dissect gene function and expression in zebrafish, with special mention of the use of these tools in cardiac biology.

## 1. Introduction

Cardiovascular diseases (CVDs) comprise heart, brain, and blood vessel diseases [1,2]. Among CVDs, coronary heart disease, cerebrovascular disease, and peripheral arterial disease are the most prevalent. Other forms of CVDs include cardiomyopathies, arrhythmias, deep vein thrombosis and pulmonary embolism, heart valve diseases, and congenital heart defects (CHDs). CHDs include a group of structural defects that arise during fetal heart development and are, thus, present at birth. These include patent ductus arteriosus, atrial septal, and ventricular septal defects [3]. CHDs arise because of inherited [4] and de novo mutations [5].

CVDs are either acquired or genetic and can be influenced by environmental factors. Acquired CVDs develop after birth, either during childhood, such as in Kawasaki disease [6] and rheumatic fever [7], or can occur later in life because of aging and lifestyle disorders. Inherited monogenic CVDs, such as homozygous familial hypercholesterolemia and hypertrophic cardiomyopathy, arise from mutations in single genes, the *low-density lipoprotein receptor* (*LDLR*) gene and *beta cardiac myosin heavy chain* gene (*βMHC*), respectively [8,9]. The discovery of single-gene mutations, such as *low-density lipoprotein* receptor (*LDLR*) mutations, with clear genotype–phenotype associations has helped to understand CVD biology and therapeutic development. Mendelian CVDs are simple to understand but form a minor class within the spectrum of clinically diagnosed cardiovascular diseases. Most inherited CVDs are polygenic and show complex traits because of underlying genetic variations and environmental factors, resulting in disease heterogeneity and variable penetrance. Such variations are difficult to map using candidate approaches.

Vertebrate animal models have been used to understand cardiac disease biology, as the cellular, molecular, and physiological aspects of human CVDs can be replicated closely in these organisms [10,11,12,13]. Efforts to create animal models of human CVDs are primarily based on the high degree of protein conservation of heart-specific genes in humans and other vertebrates, including zebrafish [14,15,16,17]. A range of animal models is available to model various physiological, cellular, and pathological aspects of CVDs, and the specific advantages and limitations are reviewed by Carlos Zaragoza et al. [10]. Zebrafish has surfaced as a prevalent model organism, offering an arsenal of genetic tools that allow the rapid in vivo analysis of vertebrate gene function and disease conditions (Figure 1). Characteristics such as a short breeding cycle, high fecundity, optically transparent embryos, the rapid development of internal organs, and easy maintenance [17,18] make this small organism popular as a research model.

Further, the zebrafish genome shows nearly 70% synteny to that of humans at the protein level, and 84% of the human disease-causing genes have zebrafish equivalent counterparts [19]. About 5 million single-nucleotide variations (SNVs) are reported in the Assam Type (ASWT) zebrafish strain [18]. Zebrafish have unique features, as an alternate vertebrate model, that offer an extensive array of paradigms to model and study human heart development and disease [20,21,22,23]. A detailed overview of the different applications of zebrafish in CVD research is illustrated in Figure 1.

In contrast to the mammalian heart, which is four chambered, the zebrafish heart is a two-chambered organ lacking a complex pulmonary vasculature. However, the early organ development and pumping function show striking similarities between fish and humans. In the zebrafish heart, the blood enters the atrium from a major vein and flows to the ventricle, which delivers it to the chief artery, as seen in the case of the human heart. The directionality of the blood flow is maintained by valves in both zebrafish and human hearts. Further, genes controlling the electrophysiological properties of human and fish hearts are conserved [24]. Zebrafish embryonic and adult heart rates, as well as the action potential (AP) shape and duration and electrocardiogram (ECG) outline, closely resemble those of humans [24]. There are a few more comparable features, such as the zebrafish heart beats spontaneously at a rate closely similar to that of the human heart. The mediolateral position of the cardiac ventricle, the existence of an atrium, and a shielding pericardial sac in the thoracic cavity are common. The *bulbus arteriosus* of zebrafish, which ensures continuous blood flow to the gill arches, is analogous to the human aortic arch, and the oxygen-depleted blood returns to the sinus venous, in fish, equivalent to the human vena cava. The zebrafish QT interval resembles that in a human electrocardiogram (ECG), with similar atrial and ventricular electrical signals [25].

In stark contrast to these resemblances, the zebrafish heart can regenerate in response to cardiac injury, a feature that the human heart lacks [26,27,28]. Zebrafish’s early survival is not completely dependent on a functional heart, as oxygen is taken up by diffusion, allowing them to survive, usually, with heart disorders for several days. This feature thus provides sufficient time for detailed analyses of the models of CVDs in zebrafish, which is impossible in human and other higher vertebrate models.

In this review, we have provided an overview of zebrafish cardiac biology and the specific contributions to CVD research using the zebrafish model by presenting an overview of the existing tools and resources available in zebrafish research. The review has been divided into three parts: The first part of this review focuses on detailed information on the morphogenetic and differentiation processes in early cardiac development. The overlap and divergence of the human heart’s genetic circuitry, structure, and physiology are emphasized wherever applicable. In the second part of the review, we overview the molecular tools and techniques available to dissect gene function and expression in zebrafish, with special mention of the use of these tools in cardiac biology. Finally, the third part explores the recent advancements in tools for zebrafish cardiovascular research.

## 2. Part I

### 2.1. Heart Development in Zebrafish: An Overview

During embryonic development, the heart is the first organ to establish in zebrafish. It constitutes different cellular types, including cardiomyocytes (CMs), endothelial cells (ECs), epicardium-derived cells (EPDCs), and smooth muscle cells (SMCs). The heart develops from two sources of mesoderm progenitors: the first heart field (FHF) and the second heart field (SHF). Anatomically, a fully functional heart development process can be divided into six distinct, yet, overlapping stages (Table 1). The earliest documented events start as early as 3.5 h post fertilization (hpf) in zebrafish [29]. The myocardial ventricular and atrial lineages are already defined at this stage. They show a specific spatial organization, with ventricular cells positioned closer to the embryonic margin and dorsal midline than atrial cells. This stage parallels a human’s first 16 developmental days (Table 1). These myocardial progenitors migrate toward the midline of the body axis and form a primitive tube by 15 hpf (early somitogenesis) as a cellular layer, with ventricular progenitors located medially relative to atrial progenitors [29]. These progenitor cells eventually fuse to form a cardiac disk by the 20-somite stage (19 hpf) [30], with an inner endocardium and an outer myocardium. At 22 hpf, this primitive disk then fuses by rotating in an anterior leftward direction to form a tube-like structure [31,32,33,34], which starts to pump the blood. The heart tube grows further by adding cardiomyocytes at the arterial and ventricular poles [35]. Finally, by 30 hpf, this tube loops upon itself to attain a chambered structure that starts ballooning or attains a curvature to increase the blood-carrying capacity. The development of the zebrafish heart is completed by 16 dpf with the formation of a valve. A detailed account of heart development in zebrafish and its comparison with that in humans and the conserved genetic circuitry are provided in Figure 2 and Table 1 and Table 2.

### 2.2. Contributions of the First and Second Heart Fields in Vertebrate and Zebrafish Heart Development

In advanced vertebrates and humans, chamber formation is gradual and progresses stepwise from two spatiotemporally distinct populations of progenitors in the heart, the first heart field (*FHF*) and second heart field (*SHF*). The *FHF* is specified and undergoes differentiation in the lateral plate mesoderm (LPM) to form the early heart tube, finally giving rise to the left ventricle. The *SHF* cells are the major structural contributors and arise in the pharyngeal mesoderm later in development. SHF cells are added to the primitive heart tube for elongation, contribute to the outflow tract, and form the right ventricle. Structurally, the zebrafish heart has four subdivisions: a, the *sinus venosus*; b, the posterior atrium; c, the anterior ventricle; and d, the *bulbus arteriosus* [31]. In zebrafish, heart development proceeds via an additive process similar to that in higher vertebrates and involves first the FHF progenitors and later SHF-specific cells that contribute to the morphogenesis of a two-chambered heart [35,47].

#### 2.2.1. Early Blastula and Gastrula

The first step in constructing the zebrafish heart involves arranging cardiac progenitor cells before gastrulation begins. Early-lineage-tracing and fate-mapping studies have shown that cardiac progenitor cells in early blastula (256- and 512-cell-stage embryos) are located in the margin between 90 and 270 degrees longitude. Ventral cells have the highest potential to become cardiac cells, and this predisposition declines in the blastomeres located laterally and toward the animal poles. This region represents the heart field in early blastula. Ventral blastomeres give rise to multiple lineages—myocardial and endothelial, including endocardial and blood cells. Notably, the atrial and ventricular chamber-specific lineages are demarcated only at the midblastula stage (2000 cells) [32]. In early blastula, a gradient determines the blastomeres’ predisposition to acquire a cardiogenic fate. Endocardial precursors are localized in the domain that is the most ventrally oriented in the heart field, whereas myocardial precursors are found throughout the heart field [48]. Cardiac progenitor cells are situated at the first four levels of blastomeres, flanking the embryonic space on both sides of the embryo [49]. The spatial arrangement of the ventricular and atrial myocardial lineages in these stages is distinct at the blastula stage (40% epiboly) before the onset of gastrulation. The ventricular cells are more marginally located toward the dorsal midline and are specified by specific signaling cues originating from the margin [50].

Cardiac progenitor cells are the first cellular population to start moving as soon as gastrulation begins. These cells band together and form two tubular primordia by the 15-somite stage. At the 21-somite stage, endocardial cells are positioned medially between the cardiac primordia and enclosed by the fusion of the myocardial tubes to form the inner layer of the heart tube. By the 26-somite stage (22 hpf), the heart tube starts beating at 25 beats/min, but there is no chamber boundary demarcation. The heartbeat gradually increases to 90 beats/min by 24 hpf and 140 beats/min by 30 hpf, when the heart loops to the right-hand side of the embryos. Looping is completed by 36 hpf, the heart beats at a rate of 180 beats/min, and chamber boundaries are evident, with the atrium on the left and the ventricle on the right side of the embryo [31].

#### 2.2.2. Transcription Factors in Zebrafishes’ Early Cardiac Development

##### Nkx2.5

The cardiogenic specification is under the control of several cardiac transcription factors (TFs), such as *nkx2.5*, *gata5*, and *hand2*, which interact with each other and other cofactors [51]. However, this process is not managed by a single TF but rather by multiple TFs expressing temporally distinct and overlapping expression domains in the cardiomyocyte precursors. By the 14-somite stage, the manifestation of the *cardiac differentiation indicator cardiac myosin light polypeptide 7 (myl7)* [52] marks the first progenitor cells denoted to become cardiomyocytes in the ALPM. In zebrafish, cardiac progenitor cells are marked by the expression of *nkx2.5. Nkx2.5* is a homeodomain-containing transcription factor, and its activity is regulated by sumoylation. *Nkx2.5* was discovered as a murine homolog of the *tinman* gene in Drosophila and is conserved across vertebrates. In zebrafish, *nkx2.5* expression in the ventral mesoderm marks a group of cells that later take up the fate of the cardiac precursor cells. However, it is essential to note that all the *nkx2.5*-positive cells do not represent the whole of the future myocardium [53,54,55,56]. The expression of *nkx2.5* can be seen in as early as five somites, 12 h post fertilization (hpf) in zebrafish, coinciding with 14–16 gestational days in humans. This expression is seen parallel to events such as the migration of precursors to the midline, cone formation, and heart tube formation, and it persists till 48 hpf [57]. Zebrafish *nkx2.5*-deficient embryos show morphogenetic defects, such as heart tube extension, but early myocardial differentiation is unaffected [58]. Overexpression of *nkx2.5* results in an oversized heart in zebrafish, and its downregulation in mice leads to defects in heart looping and the loss of *myosin light-chain 2V* gene (*Mlc2*) expression [59]. In the human fetus (stage 9 embryo) *nkx2.5* expression can be seen as a two-cell-thick layer in the developing heart tube [60]. Human *nkx2.5* mutations can cause congenital cardiac septation defects [61] and other diverse CHD anomalies. *Nkx2.5* is the primary marker of cardiac progenitors in the gastrula heart field but cannot convey cardiogenic differentiation alone and is controlled by several other transcription factors and signaling molecules.

##### Gata5

Once cardiogenesis is initiated, precardiac cells undergo further differentiation, and gata factors are typically important players in this regard. Apart from other tissues, *gata4, gata5*, and *gata6* are predominantly expressed during embryonic heart development. *Gata* factors are zinc finger transcription factors that initiate the differentiation of precardiac cells via a cardiac-restricted transcriptional program [62]. Unlike chick embryos, zebrafish embryos can develop a heart even without an endoderm, which provides an advantage for identifying the cardiogenic role of gata5 without complications arising from its additional roles in the endoderm. *Gata5* controls the number of myocardial precursors and their differentiation within the cardiac mesoderm. When overexpressed, it produces ectopic *nkx2.5* expression and converts non-myocardial cells into myocardial fate cells [56]. The zebrafish *faust* locus encodes *gata5*. In *fau* mutants at 18.5 hpf, RA myocardial precursors fail to migrate to the embryonic midline. *Nkx2.5* expression is compromised early in somitogenesis in fau mutants, suggesting an early role of gata5 in myocardial differentiation. In zebrafish, *gata5* transcripts are first observed at the dome stage (late blastula) in the most marginal cells, which give rise to endodermal progenitors and some mesodermal progenitors, as well as in the yolk syncytial layer (YSL). During gastrulation, these transcripts are further expressed [42]. Later, *gata5* expression is retained throughout somitogenesis in the LPM (10–24 hpf). By 20–22 hpf, the cardiac expression of gata5 becomes restricted to endocardial cells [56]. Zebrafish *gata5* can rescue mutants in *bmp* and *egf-cfc* signaling pathways involved in cardiac myocyte formation [57]. *Gata* factors interact with other cardiac-specific transcription factors, such as such as *nkx2.5* [58,59] *srf* [60], *mef2* [61], *tbx5* [62], and FOG proteins [63,64], and, thus, serve as master regulators in early cardiac development. In zebrafish, *gata5* induces and regulates *nkx2.5* expression in the ventral mesoderm. In contrast to *nkx2.5*, the *gata* factor function has been well conserved. For example, the fly *gata4* homolog pannier is required for the normal proliferation of cardiogenic precursors [65]. Human *gata5* maps to 20q13.2-q13, and mutations in *gata* genes may likely cause human congenital heart defects. Mutations in *gata5* lead to the tetralogy of Fallot in humans. A mouse *gata5* null allele results in a bicuspid aortic valve (BAV), a common congenital heart defect [66], and rare non-synonymous sequence variants in *gata5* are reported to be associated with human BAVs [67]. *Gata5* missense mutations have been associated with familial atrial fibrillation [68] and familial dilated cardiomyopathy [69]. *Gata5* is regulated by *bmp4* and *nodal* signaling [56,57].

##### Hand2

*Heart and neural crest derivative-expressed protein 2* (*hand 2*), a *basic helix*–*loop*–*helix* (*bHLH*) transcription factor, is expressed at the end of gastrulation in zebrafish. *Hand2* regulates myocardial specification and promotes ventricular cardiomyocyte expansion. The precardiac mesoderm marked by *nkx 2.5*- forms in hands-off zebrafish mutants but the number of cardiac myosin light chain2 (*cmlc2)* expressing cardiomyocyte precursors is reduced, and the myocardial tissue is incompletely differentiated with the absence of the *tbx5 *expression [32]. *Hand2* regulates the manifestation of *tbx5,* a T-box transcription factor essential for heart and pectoral fin growth and is mutated in Holt–Oram syndrome in humans [70,71]. In addition to transcription factors, specific co-factors, such as Pbx proteins [72], play a role in heart development. Pbx homeodomain proteins interact with *hand2* and play roles in cardiac differentiation and morphogenesis [63]. *Hand2* is regulated by *miR-1* in zebrafish, and surplus *miR-1* in the developing heart leads to a reduced pool of proliferating ventricular cardiomyocytes [64]. *Hand2* overexpression in the early stages of cardiac development in zebrafish embryos results in an enlarged heart and a larger outflow tract. Enhanced levels of *hand2* transcripts induce a corrective response to injury, suggesting its roles in cardiomyocyte production and regeneration [65]. *Hand2* mutations have been associated with congenital heart disease [66]. The *hand2* gene resides at 4q32.2-q34.3 in the long arm of chromosome 4 in humans and is associated with chromosome 4q deletion syndrome, which manifests as several forms of CHD along with other phenotypic anomalies [67]. An overdosage of *hand2* causes limb and heart deformities in a mouse model of partial trisomy distal 4q (denoted as 4q+), consistent with those observed in patients with 4q+ chromosomal disorder [68].

Cardiogenic TFs, such as *nkx2.5/csx* and *hand2*, which were identified based on sequence conservation, are expected to have conserved functions. As one key example, mouse *nkx2.5*, when appended to the Drosophila-specific tinman N-terminal domain, corrects tinman-related phenotypes in Drosophila [69]. Vertebrate *nkx* counterparts show the partial or differential rescue of the *tinman* phenotype [70], suggesting consequent divergence to attain distinct or possibly different roles in vertebrate cardiogenesis. This is evident in mouse mutants lacking *nkx2.5*, which do not exhibit the typical flies’ ‘heartless’ (tinman) phenotype but instead highlight roles for *nkx2.5* in the differentiation and morphogenesis of cardiac progenitor cells [71], myocardium specification, and looping [59] and in maintaining the ventricular gene expression program [72]. Cardiogenic TFs are functionally redundant; thus, single mutations abolishing complete cardiomyogenesis are not reported [73]. For instance, mouse *gata4/5* and zebrafish *gata4/5/6* genes have redundant roles in cardiomyogenesis. Zebrafish embryos are mutants for any single *gata4/5/6* transcription factor, and although they develop a cardiac morphogenetic defect, cardiomyocyte specification and differentiation events are not fully compromised because of the compensatory effects of the existing paralogs [74]. *Nkx2.5* also plays a redundant role with another related transcription factor, *nkx2.7*, in the process of a heart tube extension [72,75]. Transcription factors and their binding partners may form multi-protein complexes, such as *gata4, tbx5*, and *nkx2.5*, to regulate cardiac-specific genes and are implicated in human congenital heart malformation [76]. Thus, an intensive genetic analysis of transcription factors and their targets will help to better understand the process of early cardiogenic differentiation.

**Table 2 cells-14-00531-t002:** Transcription factors in zebrafish and mammalian cardiac development.

Transcription Factor	Role in Zebrafish Heart	Role in Mammalian Heart	Expression in Heart
*GATA4/GATA5* */GATA6*	Mesoderm-to-cardiac fate transition [77,78]	Anterior lateral plate mesoderm (ALPM) at early segmentation phases [78], outflow tract development [79,80]	Yes, mesoderm stage
* NKX2.5 *	Chamber formation and differentiation [47,55,81,82,83]	Differentiation of outflow tract and right ventricle from progenitors of the second heart field (SHF) [55,84]	Yes, gastrulation stage
*TBX* (T-box transcription factor family)	Cardiac progenitor formation, chamber formation and differentiation [85,86]	Cardiovascular development, homeostasis, and cardiac remodeling [85,87,88,89]	Yes, mesoderm phase
*SRF* (Serum response factor)	Cardiac crescent formation in embryos [90]	Mesoderm formation and cardiac crescent formation [91]	Yes, mesoderm phase
* HAND2 *	Promotes cardiomyocyte formation and cardiac fusion [65,92]	Formation of the right ventricle and the outflow tract in the second heart field [93]	
* MEF2 *	Drives cardiomyocyte differentiation [94,95]	Cardiac differentiation and cardiac organogenesis [95,96]	
Wt1a and Wt1b (Wilm’s tumor)	Help in transdifferentiation in cardiomyocytes, yielding epicardial-like cells [97]	Formation of the vasculature of the heart [98,99,100]	Expressed in the pericardium
TCF21	Epicardium development [101]	Cellular differentiation and cellular fate specificity [102]	Mesoderm

### 2.3. Cellular Signaling in Cardiac Development

Several signaling pathways orchestrate stepwise cellular fate decisions and determine cardiomyocyte lineage and tissue specificity.

#### 2.3.1. Hedgehog (Hh) Signaling

Cellular *Hh* signaling acts autonomously and is required continuously through gastrulation and somitogenesis, connecting the germ ring and the 10-somite stage to ensure the normal number of cardiomyocytes for both the ventricle and atrial chambers [103]. *Hh* signaling is needed, starting from the tailbud until the 15-somite stage (10–16.5 hpf) for the specification and differentiation of endocardial progenitors [104]. A knockdown of shh and *twhh*, the two zebrafish sonic hedgehog homologs, and the Hh coreceptor, *smoothened* (*smo*), results in the Nfatc endocardial marker’s expression loss.

#### 2.3.2. Fibroblast Growth Factor (FGF) Signaling

The heart size and chamber balance are fundamental to cardiac development. In zebrafish, a set of signaling cues determines the quantity of cells in both of the two main chambers, the ventricle and the atrium. FGF signaling can establish the heart dimensions and relative chamber proportion during embryonic growth. *FGFs* are secreted polypeptides that mediate cellular signaling through receptor tyrosine kinases [105]. *FGF* signaling is essential in cardiac specification [106], as seen from studies in chicks, where the endodermal *Fgf8* signal induces the expressions of *nkx2.5* and *Mef2c* [107] in mice, where *Fgf8^−/−^* [108] and *Fgfr1^−/−^* [109] mutant mice died during gastrulation. The heart size and chamber balance are both altered in zebrafish *fgf8 (ace)* mutants [110,111].

In zebrafish embryos, *fgf8* is expressed at the blastula stage all around the margin of the embryo [112], and its expression is either close to or overlaps with the cells expressing cardiac markers. From the 3-somite stage onward, *fgf8*-positive cells can be detected in the LPM; by the 4-somite stage, fgf8 expresses close to the gata4 expression domain and overlaps partially with *gata4* at the 11-somite stage. *Fgf8* expression overlaps with *nkx2.5* incompletely at the 7-somite stage and completely at the 11-somite stage. *Fgf8* expression is further seen in the ring-like structure formed by the fusion of the bilateral myocardial primordia at the 19-somite stage, and by 36 hpf, the expression becomes strongly restricted to the ventricle.

*Ace* mutant embryos show downregulated expressions of *nkx2.5* and *gata4*, suggesting a role for *fgf8* in developing cardiac precursors by initiating cardiac-specific gene expression. Further, *ace* mutants have a reduced ventricle, indicating a later role of *fgf8* in chamber development. *FGF* signaling thus has a temporal and recurring role. During gastrulation, it establishes properly sized and proportioned cardiac progenitor pools; at later stages, it becomes chamber-specific and regulates the ventricular cellular number [110]. *FGFs* activate the *RAS/mitogen-activated protein kinase (MAPK)* signaling pathway via *ETS transcription* factors and the simultaneous knockdown of three Pea3 ETS proteins—etv5, erm, and pea3, phenocopies of the defects seen in *ace* mutants [113].

The modulation of *fgf* signaling by a small-molecule antagonist of *dual specificity phosphatase* (*Dusp*6) increases the heart field size, resulting in an enlarged heart, suggesting that *Dusp6* functions as an attenuator of FGF signaling in the cardiac field to regulate the heart organ size [114]. The *Six1/Eya1* transcription complex regulates cardiovascular and craniofacial development. Mutations in mice, for both *Six1* and *Eya1*, show craniofacial abnormalities and malformations of the cardiac outflow tract and aortic arch, as seen in human del22q11, which causes DiGeorge syndrome (DGS) and velocardiofacial syndrome (VCFS). *Fgf* is a direct downstream effector of *Six1* and *Eya1* and genetically interacts with the complex. The detailed study of the *Tbx1-Six1/Eya1-Fgf8* genetic pathway may provide mechanistic insights into developing human del22q11 syndromes [115].

In humans, *FGF8* maps to human chromosome 10q25-q26 [116], and *FGF8* mutations cause the VATER/VACTERL association, which manifests vertebral defects, anorectal deformities, cardiac defects, tracheoesophageal fistula with or without esophageal atresia, renal malformations, and limb defects in several combinations [117]. *FGF8-*deficient mouse embryos show defects in the cardiac outflow tract, great vessels, and heart, coherent with the full array of cardiovascular phenotypes observed in human loss-of-22q11 syndromes [118].

#### 2.3.3. Bone Morphogenetic Protein (BMP) Signaling

Although FGF8 affects ventricle progenitors, *BMP* (*bone morphogenetic protein*) signaling is active in establishing the chamber proportions, which is prominent in imparting cardiomyocytes an atrial identity [119]. *BMPs* are a group of signaling molecules linked to the *transforming growth factor-β* (*TGF-β*) superfamily of proteins. The zebrafish mutant *swirl* revealed the roles of *bmp2b* (a BMP pathway) in the specification and differentiation of cardiac precursors [120]. Defects in the zebrafish *type I BMP receptor* gene (*acvr1l/alk8*) result in a mutation called *lost-a-fin* (*laf*), which shows a marked reduction in the number of atrial cardiomyocytes [121]. Maternal zygotic (MZ) *alk8* mutants show severe mesodermal-patterning defects and die between 16 and 24 hpf, whereas zygotic (Z) *laf/alk8* mutant embryos have a shorter ventral tailfin and a smaller atrium at 48 hpf [119,121,122]. A study described the temporal activation of *bmp* signaling in zebrafish and found that *bmp* signaling is predominantly active during gastrulation (6 hpf), and its inhibition post gastrulation did not reveal any mandatory requirement in cardiomyocyte generation [119]. Interestingly, another study in zebrafish showed that *alk8* action after gastrulation at 16 hpf (the 14-somite stage) in *laf/alk8* mutant embryos still restored the number of atrial cardiomyocytes to the same as in wild type embryos [123]. *Bmp* signaling is required for cardiomyocyte differentiation in both chambers but at different time points, with the ventricle receiving cues from gastrulation until the early segmentation stages (bud stage) and the atrium from early through late segmentation stages (12–15 somites). *Bmp receptor 1a* (*alk3a; alk3b)* double-mutant embryos lack differentiated cardiomyocytes [124]. *Bmp* signaling precedes and is required for proper cardiomyocyte differentiation. In differentiating cardiomyocytes, BMP signaling regulates myocardial differentiation by inducing the expressions of *tbx20* and *tbx2b* at mid-somite stages while simultaneously undergoing inhibition via smad6 [123]. *Bmp4* and its receptor, *Acvr1l*, are required for proepicardial organ morphogenesis in zebrafish, which gives rise to the epicardium, cardiac fibroblasts, and coronary vessels [44] and might have a pertinent role in cardiac regeneration via the activation of the adult epicardium [125].

A substitution that lies close to the nucleotide-binding site of the kinase domain of the *ALK2* receptor has been identified in human Down syndrome (DS) patients. Observing the functional activity of the p.His286Asp variant in vitro and in vivo using zebrafish suggests that the downregulation of *BMP* signaling in a DS background may result in CHD [126]. A study identified SNPs in the BMP receptor, *ALK2*; the resulting heterozygous missense substitutions were analyzed in vivo in zebrafish embryos. When injected into zebrafish embryos, the *ALK2* variant L343P RNA led to improper atrioventricular canal formation [127].

#### 2.3.4. Retinoic Acid (RA) Signaling

*Retinoic acid* (*RA*) signaling restricts the size of the cardiac progenitor population during gastrulation (75% epiboly, 8 hpf) [128]. *RA* signaling is essential to promote atrial cellular identity within the heart field and to balance the number of atrial and ventricular cells [129,130,131]. A reduction in RA signaling results in an increased density of cardiac progenitors. The zebrafish mutation *neckless* (*nls*) affects the *retinaldehyde dehydrogenase 2* gene (*raldh2*), which is expressed throughout embryogenesis and has a rate-limiting function in RA synthesis [132]. In *nls* mutants, a number of cardiac progenitor cells within the ALPM were found to be inversely proportional to the level of *RA* signaling, and a reduction in *RA* signaling initiates a fate transformation that generates an excess of myocardial progenitors at the expense of another progenitor population in the LPM [128]. Reduced *RA* signaling increases atrial and ventricle cells in different temporal windows by involving separate mechanisms [133]. RA-responsive genes, such as *Hoxb5b*, are located in the forelimb field posterior to the heart field, and *RA* signaling restricts the number of atrial cells in the heart field within the LPM through a non-cell-autonomous mechanism mediated by *Hoxb5b* [133]. The loss of the *hoxb5b* knockdown does not fully reproduce the defects observed with zebrafish embryos lacking RA signaling. Thus, the possibility of other independent signals and transcription factors being involved downstream of RA signaling in coordinating forelimb and cardiac development is to be explored. Decreased *Fgf8a* signaling can correct defects in heart and forelimb formation in *RA*-signaling-deficient embryos and has a cell-autonomous role in cardiac cellular specification but non-autonomously restricts forelimb specification. Thus, *fgf8* acts downstream of RA signaling to specify interactions between the cardiac and forelimb fields [134].

However, other independent mechanisms influence ventricular and atrial chamber progenitors. One such mode is *Cyp26*-mediated metabolic degradation, which is essential for maintaining ideal levels of RA. The loss of Cyp enzymes is reported to result in a spectrum of human cardiovascular defects. In zebrafish, *cyp26a1* and *cyp26c1* enzymes are expressed in the ALPM. The *cyp26a1*: *cyp26c1* double mutant has increased atrial progenitors in the region that normally gives rise to vascular progenitors. Thus, *Cyp26* enzymes, via *RA* signaling, demarcate the rise in the myocardial and endothelial lineage [135]. Exploring *Cyp26* metabolism in zebrafish further can expose crucial pieces of information on *Cyp26/RA-*related human anomalies. Furthermore, RA is a critical regulator of cardiogenesis, playing pivotal roles in establishing anteroposterior polarity, forming inflow and outflow tracts, and promoting the growth of the ventricular compact wall. *RA* signaling has been implicated in multiple developmental processes, including the specification of cardiomyocyte subtypes, epicardial development, outflow tract formation, and coronary arteriogenesis [136]. Additionally, previous studies investigating *Raldh2* expression and its effects on *FGF* signaling have provided insights into the spatiotemporal requirements of *RA* in cardiac development and repair [137]. These findings highlight *RA’s* dual role in cardiac morphogenesis and its potential regenerative effects, which warrant further exploration in the context of cardiac development.

#### 2.3.5. Wnt Signaling

*Wnt* signaling plays an important role in the cardiac progenitors specified during the induction of the ventrolateral mesoderm [138,139]. However, in the precardiac mesoderm, *Wnt* signaling acts temporally and plays different and distinct roles throughout vertebrate cardiac development [138,139,140,141]. During early zebrafish cardiac development, increased *Wnt* signaling promotes *nkx2.5* expression, which marks the precardiac mesoderm at the pre-gastrula stage up to 5 hpf. In contrast, it inhibits cardiac specification once gastrulation begins and later during somitogenesis [139]. The downregulation of *Wnt/βcatenin* signaling is mandatory to guide the mesoderm to the cardiovascular lineages for the formation of bilateral heart fields in the ventrolateral mesoderm [139,142]. Thus, positive *Wnt* signals are required early for heart differentiation, and in later stages, once gastrulation begins, the signals become negative to restrict the heart field.

*Wnt* signaling is required for establishing the proper number of atrial cells, and it is suggested that the inhibitory effect of *Wnt* signaling on heart formation during gastrulation could be because of the induction of apoptosis in cardiomyocyte progenitors [143]. *Wnt* signaling is further regulated by certain proteins, such as *TMEM88* (target protein transmembrane 88), a two-transmembrane-type protein that acts as a negative regulator of Wnt [144]. The zebrafish *tmem88a* is expressed mainly in the lateral plate mesoderm, including the bilateral heart fields.

The knockdown of the zebrafish *tmem88a* activates the *Wnt* pathway, resulting in cardiomyocyte deficiency in zebrafish embryos, whereas *Wnt* inhibition promotes progenitor specification and cardiogenesis. *Tmem88a* acts downstream of the *gata5/6 gene* in the precardiac mesoderm to impede *Wnt* signaling and expand the number of cardiac progenitors [145]. *Tmem88* bears a chromatin signature unique to genes that mediate cellular fate decisions [146]. The *tmem88* knockdown does not affect the number of myocardial progenitors, as *nkx2.5,* or *gata4*, expression is maintained; however, the expressions of genes that are highly expressed during cardiomyocyte development are significantly diminished, suggesting alterations in cellular fate programs. TMEM88 is, thus, essential for restricting cellular fate decisions in the precardiac mesoderm to mark the onset of cardiomyocyte specification by inhibiting *Wnt* signaling [146]. The effects of *Wnt* signaling on early cardiomyocyte formation can be easily translated to the embryonic stem (ES) cellular culture, underscoring the potential for creating novel stem-cell-based therapies to target congenital heart defects and cardiomyopathies [147,148].

A few lacunas limit the understanding of cellular signaling pathways in early cardiac development. The current perception of the key combinatorial signals that determine the correct distribution and number of cardiac progenitors at defined time points during cardiac development is constrained. Further, the gene regulatory networks downstream of the signaling pathways across species might be diverged and are not fully uncovered. Extracellular signals are involved in multiple cross-talks and exert many inter-inhibitory and modulatory effects [149,150,151]. The present insight into these mechanisms involving multiple players in this context precludes our understanding of myocardial differentiation and morphogenesis. Signaling ligands and their receptors can be conditionally expressed spatiotemporally across developmental time points in zebrafish. The pharmacological inhibition or activation of these pathways in vivo can be easily performed in zebrafish. This may generate models that depict interactions among the major cardiogenic pathways. Such efforts may be the premise for creating and utilizing information on cardiac myogenesis for therapeutic interventions.

#### 2.3.6. Heart Looping: Chamber Formation

One of the most crucial steps, the event of cardiac looping, is a conserved hallmark of heart development. The key feature of cardiac looping is the loss of bilateral symmetry, and the heart becomes a helically coiled loop that gains a left–right asymmetry [152]. Laterality defects give rise to morbid conditions often associated with CHDs [153]. For the heart tube to loop, the ventricle starts looping and bends toward the right of the midline. This leads to torsion at the looping axis, which initiates physical cues for the formation of the heart valve or atrioventricular canal (AVC).

In zebrafish, laterality signals from the Kupffer’s vesicle, a ciliated region, establish asymmetry in the heart tube, in the direction of ‘heart jogging’ and that of ‘heart looping’ [154]. ‘Heart jogging’ refers to a process by which the heart tube is displaced relative to the dorsal midline with a leftward ‘jog’ [155]. Cardiac jogging occurs at 24 hpf before looping starts such that the venous pole is aligned leftward, and the arterial pole stays at the midline of the body’s axis [156]. The heart finally becomes asymmetric by looping to the right side by 36 hpf; the direction of the cardiac jogging influences the direction of the looping [156], and the cardiac cone undergoes clockwise rotation during tubular formation so that left–right symmetry changes to dorsal–ventral polarity [152]. Once looping has occurred, bulges start to appear in the heart tube, and the chambers take their shape. This process of chamber ballooning is another essential step in the creation of a fully operational heart. During chamber ballooning, the lumen of the ventricle and the atrium increases, which in turn helps the cardiac walls to attain a proper shape and gives strength to pump the blood.

The advection forces generated by the pumping blood provide the physical cues for chamber ballooning. In the ventricle, these morphogenetic changes are governed by changes in the cellular shape [157]. Cardiac contraction forces seem to control these cellular shape changes, as demonstrated in zebrafish mutants, such as the *weak atrium* (*wea/myh6*) [157], *half-hearted* (*haf/vmhc*), and *silent heart* (*sih*) mutants [158]. Rigor provided by the blood flow helps the ventricle cells to undergo elongation and reorientation, whereby the ventricle cells elongate and lie perpendicular to the atrioventricular axis. The change in the shape (elongation) and reorientation of the ventricular myocardial cells is very important for providing strength to the ventricular walls.

Nodal, the human ortholog for the zebrafish *southpaw* is an embryo-patterning gene and functions to establish left–right patterning in various organ systems, such as the left–right patterning during heart tube looping [159,160]. Nodal mutations in humans lead to looping defects [161]. Nodal signaling directs cardiac cellular migration by enhancing cellular velocities on the left of the cardiac cone, whereas the *bmp* pathway adversely controls cardiac cellular migration rates. These differences in cellular velocities within the L/R axis lead to the rotation of the tube to form an asymmetrically jogged heart. *Nodal* signaling is mandatory for this process, whereas *bmp* is not. However, when nodal signaling is reduced, heart cells’ responses to Bmp4 signals are enhanced, suggesting a combinatorial synergy between the two *TGFβ* pathways. Genetic analysis has revealed that transcription factor FoxH1 integrates both Nodal and Bmp signals to establish proper cardiac laterality [162]. The process of heart looping appears to be conserved from zebrafish to humans [163]. In contrast, the process of heart jogging is unique to zebrafish [156] and does not occur in mammals, except for a less similar leftward movement in the mouse heart, which might be analogous to heart jogging in zebrafish [164]. The zebrafish *tbx1*^-/-^ mutant shows cardiac defects, such as conotruncal and ventricular septal defects, similar to those seen in DiGeorge syndrome (DGS) patients [165]. The expressions of both Wnt11r and *alcama* (*activated leukocyte cell adhesion molecule a*) are downregulated in *tbx1-/-* mutants, and both *Wnt11r-/-* mutants and *alcama* morphants have heart looping and differentiation defects similar to those in *tbx1*^-/-^ mutants. The ectopic expression of *wnt11r* or *alcama* partially rescues these defects. *Tbx1,* thus regulates heart looping and differentiation in a linear pathway through *Wnt11r* and *alcama* [165].

The roles of noncoding RNAs in left–right symmetry and cardiac development have been described for zebrafish; miR-19b has been shown to regulate laterality development and heart looping in zebrafish embryos by targeting *ctnnb1* and, thus, inhibiting *wnt* signaling [166], and miRNAs are known to play roles in chamber ballooning. One example is miRNA-143, which regulates *Adducin3*, an F-actin capping protein. *Adducin3* plays roles in cellular differentiation and elongation and, when downregulated by miRNA-143, affects chamber morphogenesis because of defects in cytoskeletal polymerization [167]. Research on how the structure–function relationships bring about localized changes in the cellular shape and size remains limited. Zebrafish offer an in vivo system to study the ballooning myocardium to understand how cardiac morphogenesis is guided by conductive and contractive forces. Detailed accounts of zebrafish heart development have been described by Scott et al., 2010 [168]; Bakker et al., 2011 [34]; Liu et al., 2012 [169]; and Francoeur N. et al., 2021 [170].

## 3. Part II

### 3.1. Zebrafish Cardiovascular Toolkit

An in-depth understanding of biology also stresses technological developments to explore and validate the available information and further create new additions to the existing repertoire of knowledge. Table 3 provides a comprehensive summary of information on zebrafish resources. We discuss some of the established tools, such as transgenesis, high-resolution imaging, gene knockdown, and genome editing, that have been developed in zebrafish to explore the biological landscape of this organism. We also summarize the significant contributions of these tools in cardiac biology.

#### 3.1.1. Transgenesis to Study Zebrafish Cardiac Development

In the late 1980s, as zebrafish emerged as an alternate vertebrate model organism for studying development and genetics, several focused efforts were initiated to establish methods and vectors for transgenesis. The earliest methods included the microinjection of bacterial plasmid DNA [189,190] and pseudotyped retroviruses into zebrafish embryos [191]. Although these approaches could successfully yield stable transgenics, they suffered from the major limitation of low efficiency and unreliable expressions. The pseudotyped retroviruses were further improved for their efficiency in germline transmission and cloning genes using insertional mutagenesis [192]. The field further advanced with the use of GFP as a reporter in live zebrafish embryos [192,193]. However, the need for a robust transposon-based insertional mutagenesis system, such as the P element of Drosophila, remained unrealized in zebrafish. The discovery of the tol2 system, from *Oryzias latipes *(medaka) revolutionized transgenesis in zebrafish [194,195,196,197,198,199,200,201]. It is, so far, the most efficient system for transgenesis across vertebrates, including zebrafish. The tol2 toolkit consists of various insertional vectors that can be used to create fluorescent reporter lines [197,198] and monitor spatiotemporal gene expression. Transgenic zebrafish fluorescent reporter lines have been created to aid the visualization of the cardiovascular system in real time (Figure 3). The pan-endothelial *Tg (fli: GFP)* or *Tg (tie2: GFP)* transgenic lines have proved to be extremely useful in precisely tracking the development of the cardiac endothelium [202,203]. Cardiomyocyte-specific transgenic lines *Tg* (*cmlc2: GFP*) [204] and *Tg (cmlc2: ras-GFP)* [205] have been used to study events such as cardiac trabeculation [44]. Similarly, *Tg (cmlc2: gCaMP)* has been utilized for studying cardiac contraction using calcium excitation as the readout [23,206]. The ease of generating transgenic zebrafish expressing reporter genes under the control of endothelial-specific promoters reduces the need for perfusion. For example, *flk1: GFP*, in transgenic zebrafish expressing GFP, localized to the endothelial cellular nucleus [207], whereas *fli1: eGFP*, in transgenic zebrafish, expresses GFP in the entire endothelial cytoplasm. Crossing these with the *gata1: dsRed* transgenic line that labels erythrocytes enables the assessment of both the blood flow and vascular anatomy in a single embryo. In this area, previous studies by Missinato et al. (2021) have utilized the transgenic zebrafish *(Tg (myl7: fapdl5-cerulean)),* and have shown that zebrafish can regenerate adult cardiac tissue following injuries from ventricular apex amputation, cryoinjury, and cardiomyocyte genetic ablation [208]. However, chemoptogenetic cardiomyocyte ablation leads to a distinct regenerative process. These results suggest that different cardiac injury models engage distinct regenerative mechanisms, and chemoptogenetic ablation provides a useful system to study cardiomyocyte-specific injury, proliferation, and regeneration in isolation from broader tissue responses and whole-organ activation [208].

A collection of CET (cardiac enhancer trap) lines expressing GFP in all the major cardiac cellular types, viz., the endocardium, myocardium, and epicardium and cardiac chambers, is available for in vivo studies of heart development, physiology, and drug screening [209]. The Casper (*Tg *(*cmlc2: nuDsRed*)), which has a red fluorescent heart, has been used for quantifying cardiac functions in vivo in zebrafish hearts [206]. The application of transgenesis in developing the UAS-Gal4 system for targeted gene expression in a spatiotemporal manner in zebrafish is noteworthy. This binary expression system works with two lines—an activator line and an effector line. The activator line expresses the yeast transcription factor GAL4 under tissue- or stage-specific promoters, while the effector line expresses the target gene under upstream-activating-sequence (UAS) regulation. The two lines are crossed, and specific expressions or misexpressions can be studied in the progeny. Currently, a resource consisting of various transgenic fish expressing Gal4 in particular cells is available to facilitate the visualization of specific cells in vivo by targeted gene expression [210,211].

A recent study describes the *Gal4 UAS* system being utilized to create a genetically encoded, optically controlled pacemaker by expressing light-gated ion channels, halorhodopsin, and channel rhodopsin in zebrafish cardiomyocytes. The pacemaker’s location could be easily traced via illuminating the zebrafish heart at 3 days post fertilization (dpf). This transgenic line was used to stimulate reversible tachycardia, bradycardia, atrioventricular blocks, and cardiac arrest and, hence, is a good tool for understanding the emergence of pacemaker activity during early cardiac development [212]. Besides this, *Gal4*-based gene traps have also been utilized for gene disruption in zebrafish [213].

The application of transgenic fluorescent lines has far revolutionized the technology itself. Transgenic fluorescent reporter lines have been used to study organ/tissue development, analyze gene expressions, analyze cellular transplants, track cellular lineages and their migrations, analyze tissue-specific gene regulatory elements, sort labeled cellular types, and screen mutagenesis and chemicals. Thus, the zebrafish model has a transgenic arsenal for investigating the cardiovascular landscape at molecular and cellular levels. Leveraging its transparency, genetic tractability, and suitability for high-resolution imaging, Rajpurohit et al. introduced a transparent transgenic zebrafish model using the Casper mutant strain (*roy−/−, nacre−/−*) coupled with cardiomyocyte-specific fluorescent markers [173]. This model allowed the unprecedented real-time visualization of apoptotic and necrotic pathways in cardiomyocytes, significantly advancing our ability to study cardiac injury, regeneration, and potential therapeutic interventions [173]. Complementarily, the development of the *Tg(myl7: GA)* transgenic line (which employs GFP–aequorin as a bioluminescent calcium indicator) extended zebrafish utility by facilitating the continuous noninvasive monitoring of calcium dynamics in live larvae over extended periods. This surmounted limitations inherent in fluorescence-based imaging techniques, such as photobleaching and motion artifacts [174].

Various studies (including both studies previously exploring the development of transgenic zebrafish lines) share a focus on the zebrafish’s intrinsic advantages, particularly its transparency and genetic tractability, which enable sophisticated manipulations and detailed investigations of cardiovascular physiology and pathology [173,174,214]. Rajpurohit et al.’s model provides unique insights into cardiomyocyte injury responses and cardiac regeneration, while the *Tg(myl7: GA)* line offers a sensitive platform for analyzing calcium signaling (a central component of cardiac functions [173,174]. Through pharmacological studies, the *Tg(myl7: GA)* model demonstrated the precise detection of calcium flux alterations induced by agents like Bay K8644 and nifedipine and its application in modeling conditions such as terfenadine-induced heart failure [174]. These applications highlight zebrafish as a versatile model for studying physiological processes and disease states.

#### 3.1.2. Cardiovascular Imaging in Zebrafish

The transparent canvas of zebrafish embryos, coupled with high-resolution microscopy, provides a visual treatment for obtaining a clear view of internal structures/organs at a subcellular level without invasive instrumentation. In particular, the heart and blood vessels can be easily observed under a simple microscope at a resolution where an observer can quantify cardiac contraction and relaxation [215,216]. Zebrafish cardiac mutations have uncovered several genetic pathways, thus revealing the complexity of heart morphogenesis and functions [155,158]. To study in real time the molecular mechanisms and cellular rearrangements that regulate morphogenetic processes, such as cardiac ballooning or tubular assembly, high-resolution advanced imaging techniques have been developed and utilized in zebrafish [217].

The embryonic zebrafish heart undergoes rhythmic, successive contractions of the ventricle and atrium to drive blood circulation [215]. The basic contractile unit in a cardiomyocyte is the sarcomere, which undergoes constant cellular repolarization and depolarization by releasing Ca^2+^, Na^+^, and K^+^ ions [218]. Contractility defects are generally described qualitatively as reduced heartbeats or arrhythmias. However, the quantitative analysis of defects in the conduction system or contractile apparatus is required to assess heart function. Current high-speed imaging technologies can resolve the functional parameters of the cardiac contraction process and help to understand underlying causes, such as nonfunctional sarcomeres and disturbed calcium cycling [217].

A detailed 3D time-lapse imaging of a beating heart in zebrafish was conducted to understand the initiation of cardiac trabeculation [219,220]. Trabeculation is a cellular process involving cell–cell contacts to generate physical forces and is required to establish ventricle conduction and contraction. The imaging data revealed that cardiomyocytes physically interact with the trabecular layer via protrusions to form new cell–cell contacts just before the myocardial components invade the trabecular layer. The group further examined mutations with defective trabeculation, such as *erbb2* (no protrusions and a lack of contraction and flow), *tnnt2a *morphants (unstable protrusions), and myh6 morphants (a decreased number of protrusions and no atrial contractions), to further understand the role of cell–cell contacts in cardiac trabeculation [220].

An 80 MHz ultrahigh-frame-rate echocardiography system has been developed with Doppler-gated technology for quantifying ventricular function in zebrafish hearts [221]. This technology can precisely identify the timings of the ventricle’s end diastole (ED) and end systole (ES). In contrast to the technical difficulties of angiography in small mammals, two-dimensional “angiograms” of most vessels can be generated in zebrafish, without instrumentation, according to the “digital motion analysis” of moving blood cells [202,222,223].

The zebrafish model has been utilized to obtain shear force measurements in vivo. Shear forces generated by blood flow are the key regulators for shaping the developing heart. High-speed bright-field imaging was used to attain a time-lapse sequence of the blood flow inside the beating heart between 3 and 6 days post fertilization. The velocity of red blood cells within the heart chambers can be calculated using particle image velocimetry. The position of the heart wall could be determined by demarcating the boundary of the blood flow, thus generating the required information to measure the wall shear in vivo. The dynamics of the shear forces, within an early developing heart, across the chambers can thus be determined [224].

A high-speed imaging platform, integrating high-speed camera-based imaging (up to 1000 frames/second) and 3D-confocal-microscopy-based methods, was developed to assess vascular functions in zebrafish. This method can be used to quantify the blood flow, vessel diameter, and endothelial barrier function in real time. The method can also be used to image and record the blood flow in a large (1 mm) segment of the vessel over many cardiac cycles with sufficient speed and sensitivity. Using this method, the trajectories of individual erythrocytes can be traced, and changes in the three-dimensional sizes of vessels can be deciphered at high resolution. Such a method holds promise to be applied to zebrafish models of human vascular diseases [180].

Several in vivo imaging technologies, including particle-tracking velocimetry, micro-Doppler imaging, digital particle image velocimetry (DPIV), and four-dimensional-DPIV, have been established to study the fluid flow in developing zebrafish hearts. These methods rely primarily on the use of fluorescent tracer particles, and their application is tremendously important in understanding cardiac structure–blood flow relationships [225]. Hove JR et al. have reported standardized protocols for introducing tracer particles to the cardiovascular system of zebrafish and have conducted high-resolution blood flow mapping [226].

A high-frequency ultrasonic system, consisting of 75 MHz B-mode imaging at a spatial resolution of 25 microns and a 45 MHz pulsed-wave Doppler ultrasound, was used to image adult zebrafish hearts. All the structures of the heart, atrium, ventricle, *bulbous arteriosus*, atrioventricular valve, and bulboventricular valve can be reproduced as real-time images, and image-guided Doppler-based blood flow measurements are possible. Using this system, the E-flow and A-flow in ventricular filling and the development of diastolic flow reversal at the *bulbous arteriosus* were deduced, and the data suggested functions similar to those in higher vertebrates [181].

Some imaging advancements have contributed to understanding cellular movements during regeneration and stem cellular dynamics. PhOTO zebrafish lines provide one such example. These lines work on the principle of irreversible photoconversion in vivo and have been developed to aid lineage tracing during highly active events, including cellular movements, cancer, and tissue regeneration. Finer details, such as monitoring changes in cellular shape, cellular divisions, and dynamics, can be achieved by the specific fluorescent labeling of the plasma membrane and nuclear proteins. PhOTO zebrafish, thus, have broad applicability for lineage tracing in the early embryo as well as in the adult, in the contexts of development, regeneration, cancer progression, and stem cellular biology [227,228]. Tissue-specific luciferase-based transgenic lines were generated for the easy evaluation of regeneration and engraftment in freely moving adult zebrafish. Luciferase-based live imaging in these lines can be used to estimate muscle quantity in the heart and help to track injury-induced cardiac regeneration in single animals. Furthermore, the luciferase-based approach enables better visualization and quantification of the transplanted hematopoietic stem cell progeny, with better sensitivity and spatial resolution compared to those of fluorescence-based detection [229].

Meanwhile, many new mutations and cardiac-specific protein trap lines are being generated [200,201]. The development of imaging technologies should further spread out the field’s horizons. For example, using fluorescent protein trap lines to track cellular movements during cardiac development will be particularly exciting. High-resolution real-time imaging in the zebrafish model should help to further illuminate the knowledge of mammalian cardiogenesis and the origin of heart disorders.

#### 3.1.3. Electrophysiology: Measuring the Zebrafish Cardiac Function

Zebrafish ECGs resemble those of humans. Some essential features, such as the heartbeat rates in embryonic and adult zebrafish, the shape and duration of the action potential (AP), and other characteristics of an electrocardiogram, are closely similar to those of human electrocardiograms (Figure 4). The basal heartbeat rate of adult zebrafish is closely similar to that of humans (120–130 beats/min) at the optimal water temperature (28 °C) [24,230,231]. AP propagation in zebrafish and human hearts also reflects the similarity of the ECG morphologies [232]. As in human ECG recordings, a distinct P-wave, QRS-complex, and T-wave are observed zebrafish ECG recordings [22,233], signifying that depolarization and repolarization in the zebrafish heart are analogous to those in the human heart [232]. The average QT interval and ventricular AP duration in zebrafish are ~300 ms [234]. Thus, the QTc interval is somewhat shorter than that in humans. Nevertheless, it is important to note that zebrafish ECG and AP values are gained at lower temperatures than human ECG and AP values.

A few other measurements, such as the late rectifier K^(+)^ current (I_(Kr)_), are well understood. Therefore, studying ion currents in zebrafish myocytes may represent a valuable model for exploring human I(Kr)-channel-related ailments, including long QT syndrome [235]. Furthermore, combining optogenetic techniques and transgenesis tools has enabled studies on disturbed heart rhythms [212]. Thus, zebrafish have become a relevant research model for exploring ion-channel conditions associated with abnormal repolarization [235].

A number of methods and tools are available for cardiac measurements in zebrafish. Zebrafishes’ whole-heart electrical activities are regularly documented using in vivo electrocardiography (ECG) [232]. In addition, various noninvasive microscopic video recording methods have been developed to determine heart rates [236,237,238], quantify fractional ventricular shortening (a measure of the systolic contractile function) [239,240,241], and examine blood flow changing aspects by tracking the movement of erythrocytes or fluorescent molecules incorporated into the circulation [222,242].

Ca^2+^-sensitive fluorescent dyes [243,244,245,246] or the fluorescent Ca^2+^ indicator transgene (*Tg(cmlc2:gCaMP))* [217] can be used for the assessment of cardiac conduction and excitability, and transmembrane APs may be evaluated using voltage-sensitive dyes [247]. A thorough investigation of cardiac conduction velocities, initiation forms, and arrhythmias is possible by applying these voltage-sensitive dyes during so-called optical mapping. These high-resolution imaging methods are influential tools for the study of zebrafish cardiac physiology but require the complete absence of cardiac contraction that can be easily achieved using an excitation–contraction uncoupler, such as blebbistatin, which abolishes contractility without significantly altering the AP morphology.

The cardiac APs of zebrafish may be quickly recorded from the intact heart using micro-electrodes [24], patch–clamp technology [248], and, similarly, voltage-sensitive dyes can also be applied [247,249]. Both atrial and ventricular monophasic APs from intact adult zebrafish beating hearts at the physiological temperature of 28 °C can be recorded. Compared to human and mouse myocardia, zebrafish atria and ventricles showed substantially lower maximum AP upstroke velocities [24]. Notably, the resting membrane potential is equivalent in zebrafish and humans, suggesting that the variances in the upstroke velocity among species were not because of differences in Na+ channel disposal. In the case of zebrafish, a long-lasting plateau phase follows the AP upstroke and ends with a phase of rapid terminal repolarization [248]. A prominent plateau phase with a shorter AP is displayed by the atria and ventricles in humans [250,251]. Despite the contrast in the morphological characteristics of zebrafish myocytes and human myocytes [252], the inclusive profile of the adult zebrafish ventricular AP is equivalent to that of the human heart AP, and human APs appear to be more comparable with zebrafish APs than mouse APs [253,254,255,256].

### 3.2. Revisiting Genetic Tools in Zebrafish Cardiovascular Research

In recent years, the zebrafish model has become a cornerstone in cardiovascular research, offering a wealth of information about heart development, disease mechanisms, and regenerative processes. However, as genetic tools evolve, it is crucial to consider whether the current discussions on genetic manipulation techniques—such as reverse and forward genetics, zinc fingers, TALENs, TILLING, and CRISPR—remain relevant or whether these sections have become somewhat redundant. Many of these techniques have been extensively reviewed in the last decade, and it is essential to examine how they have evolved and what new knowledge can be garnered from them in the context of zebrafish cardiovascular research.

#### 3.2.1. Forward and Reverse Genetics in Zebrafish Models

Both forward and reverse genetics have long been foundational for understanding cardiovascular diseases in zebrafish. These methods involve systematically manipulating genes to observe the resulting phenotypic effects (forward genetics) or studying the effects of specific mutations or knockouts (reverse genetics). Although these techniques remain critical, their foundational concepts have been covered extensively in past reviews [257,258,259,260,261]. Reverse genetic tools help to study the role of a previously identified gene for which functions are unknown [262]. This approach has been effectively utilized to develop disease models in zebrafish. One of the successful methods in reverse genetics is the temporary “knockdown” of a developmentally important gene by injecting morpholino antisense oligonucleotides into the fertilized egg and monitoring the phenotype during early development [262]. Morpholinos act by blocking translation or aberrantly inducing splicing by targeting either the transcriptional start site or an internal splice site of the mRNA of the target gene [262,263]. Although these approaches have led to the identification of numerous cardiovascular genes, the excitement around them is diminishing, as their utility has become well-established and, in many cases, routine. New insights may not come solely from revisiting these methods but rather from enhancing their precision or combining them with new tools, such as single-cell sequencing or CRISPR-based technologies [264,265,266,267], which can now interrogate gene function more dynamically and comprehensively. In this direction, our group has identified a novel RNA transcript (*grin22bb ART*), which regulated calcium signaling and caused cardiomyopathy and heart failure in adult zebrafish, using the *tol2-*based gene trapping approach [268]. This model provides an avenue to study heart failure in adult zebrafish.

#### 3.2.2. Zinc Finger Nucleases (ZFNs), TALENs, and TILLING: Revisiting Traditional Approaches

The developments of zinc finger nucleases (ZFNs) [269,270,271,272], transcription activator-like effector nucleases (TALENs) [273,274,275], and targeting local lesions in the genome (TILLING) [276] have provided powerful alternatives for inducing targeted mutations and investigating gene functions in zebrafish. However, despite their early promise, these techniques have been widely surpassed by more efficient and versatile technologies, particularly CRISPR/Cas9. Although ZFNs and TALENs remain useful for precise genetic editing, their complexity and labor-intensive nature make them less appealing than CRISPR/Cas9, which has revolutionized the field by enabling high-throughput, specific gene editing with fewer off-target effects [277]. TILLING is particularly useful for identifying point mutations in large populations, but like ZFNs and TALENs, it has been eclipsed by newer genome-editing approaches that are faster and more precise. Thus, although these technologies still hold value in specific contexts, it is important to recognize that they no longer represent the cutting edge of genetic manipulation techniques. For instance, with the increasing accessibility of CRISPR/Cas9 systems and the advent of newer CRISPR-based tools (such as prime editing and base editing), the focus in zebrafish cardiovascular research is shifting toward these more advanced and streamlined approaches. This is a feasible method for identifying stable loss-of-function mutations in known genes [278,279]. TILLING enables phenotypic assessment at later stages of development than the temporary knockdown induced by morpholinos. Because TILLNG requires large mutant libraries and significant DNA-sequencing capacity to identify a desired mutation, it is labor intensive and generally not feasible for a standard laboratory.

#### 3.2.3. Clustered Regularly Interspaced Short Palindromic Repeats/Cas (CRISPR)

The type II prokaryotic CRISPR, an adaptive immune system from Streptococcus pyogenes, facilitates RNA-guided site-specific DNA cleavage [280,281]. Several type II CRISPR/Cas systems have been engineered to guide Cas9 nucleases in modifying endogenous genes within human, mouse, and cultured cells [282,283] and fly genomes [284]. In zebrafish, the CRISPR/Cas system has been shown to induce targeted genetic alterations in embryos, with efficiencies comparable to those achieved utilizing ZFNs and TALEN [281]. The CRISPR/Cas9 system shows an added advantage of efficiently inducing bi-allelic mutations in the F_0_ generation, with a mutagenesis rate of 75–99%, thus allowing for phenotype analysis directly in the injected embryos [285]. The specific Cas/gRNA has been used to target cardiac-specific genes *etsrp *and *gata4* or *gata5* in zebrafish embryos in vivo, and the bi-allelic conversion of *these genes* in the resulting somatic cells was observed. It was shown that Cas9/*gRNA*-injected embryos recapitulated their respective vessel phenotypes in etsrp (y11) mutant embryos or cardia bifida phenotypes in fau (tm236a) mutant embryos [280]. An RNA-guided Cas9 nuclease system, with a better-targeting array of RGNs from one in every one hundred twenty-eight base pairs (bps) of random DNA sequences to one in every eight base pairs and germline transmission rates in zebrafish reaching 100%, has been developed. The group also demonstrated that these nucleases and single-stranded oligodeoxynucleotides (ssODNs) together can produce single-nucleotide substitutions and other desired sequence modifications [281]. In a study by Liu et al., the application of CRISPR/Cas9 technology in zebrafish for cardiovascular research was demonstrated through innovative and high-throughput genetic-screening techniques. The study utilized pooled single-guide RNAs (sgRNAs) with Cas9 to identify transcriptional regulators critical to cardiomyocyte functions. This approach identified *zbtb16a*, a gene essential for cardiac development, with findings highlighting its pivotal roles in early heart formation and structural integrity [171]. Additionally, the study showcased the utility of multigene targeting, wherein up to five genes were simultaneously disrupted to investigate complex gene interactions, thus providing insights into the genetic networks governing heart morphology and physiology. By leveraging the efficiency of CRISPR/Cas9 and the genetic tractability of zebrafish, this work highlights the potential of zebrafish as a robust model organism for dissecting the molecular underpinnings of cardiovascular biology, offering pathways for understanding congenital heart diseases and potential therapeutic targets [171].

A study by Li et al. used a cardiomyocyte-specific *Klf15* knockout mouse model to investigate the circadian susceptibility of the myocardium to ischemia reperfusion injury, uncovering the roles of *Klf15* in regulating NAD+ levels and mitigating mitochondrial dysfunction during ischemic stress [286]. Although mouse models, such as the *Klf15* knockout used by Li et al., provide valuable insights into specific molecular pathways, like the circadian regulation of myocardial injury, zebrafish offer unparalleled advantages for studying cardiac regeneration and repair.

As these technologies advance, the potential for multiplexed gene editing in zebrafish will enable the creation of complex models for multifactorial diseases, like cardiovascular disorders. For example, multiplexing CRISPR/Cas9 systems can simultaneously target multiple genes, providing a more accurate representation of the polygenic nature of diseases. Furthermore, with the advent of CRISPR interference (CRISPRi) and CRISPR activation (CRISPRa), researchers can fine tune gene expressions rather than knockout genes entirely, allowing for more nuanced insights into the roles of genes in cardiovascular diseases [287]. These advances position CRISPR/Cas9 and its variants as integral tools for developing zebrafish models that are not only more accurate but also closer to the complexity of human disease models. Although traditional genetic tools for zebrafish, such as ZFNs, TALENs, and TILLING, have certainly improved our understanding of cardiovascular development and diseases, it is clear that the field is moving toward a new era defined by next-generation genome editing, multiomic approaches, and artificial intelligence (AI). The refinement of CRISPR/Cas9 technologies, coupled with advanced sequencing methods, such as single-cell RNA-seq and spatial transcriptomics, will likely unlock new dimensions of heart disease modeling that were previously unimaginable.

#### 3.2.4. CRISPR/Cas9 and Beyond: The Frontier of Genetic Engineering in Zebrafish

CRISPR/Cas9 has undoubtedly revolutionized the field of genetic research, including in zebrafish cardiovascular studies. The simplicity, efficiency, and versatility of CRISPR/Cas9 have made it the go-to tool for genome editing in zebrafish. This technology has been employed extensively to study gene functions, model human cardiovascular diseases, and even investigate heart regeneration mechanisms [288]. However, although CRISPR/Cas9 remains a workhorse, its applications in zebrafish research rapidly expand into prime editing and base editing, offering even greater precision and fewer off-target effects [289,290].

The potential for multiplexed gene editing in zebrafish will enable the creation of complex models for multifactorial diseases, like cardiovascular disorders. For example, multiplexing CRISPR/Cas9 systems can simultaneously target multiple genes, providing a more accurate representation of the polygenic nature of diseases. Furthermore, with the advent of CRISPR interference (CRISPRi) and CRISPR activation (CRISPRa), researchers can fine tune gene expressions rather than knockout genes entirely, allowing for more nuanced insights into the roles of genes in cardiovascular diseases [287]. These advances position CRISPR/Cas9 and its variants as integral tools for developing zebrafish models that are not only more accurate but also closer to the complexity of human disease models.

#### 3.2.5. Drug Screening in Zebrafish

Features such as the high fecundity, permeability to small molecules, and small size of zebrafish embryos make zebrafish an amenable vertebrate model for high-throughput drug and pharmacological screening similar to cell-based assays [291,292,293]. A single pair of zebrafish can produce 200–300 embryos each week. Test compounds can be added directly to the water/medium surrounding the embryos [292,294]. The short developmental time frame of zebrafish significantly condenses the time needed for experimentation. Small-molecule libraries for molecules with potential cardiovascular effects have been tested in various assays [291,295,296,297]. Molecules that target vascular endothelial growth factors to prevent aortic occlusion have been identified in such assays [298]. The specific effects of anti-angiogenic drugs have been studied using automated image analysis in transgenic zebrafish embryos expressing GFP in the endothelium [299]. An anti-angiogenic agent discovered in this screening was subsequently shown to inhibit human endothelial cell tube formation in vitro, demonstrating that positive hits from unbiased drug screening with zebrafish can translate into effects in human-cell-based assays.

Zebrafish biologists have created a method for the rapid isolation of adult zebrafish hearts and their maintenance ex vivo, along with a setup to perform fast small-molecule-throughput screening assisted by an in-house-implemented analysis script. The fact that a range of readouts can be acquired along with the in-house-developed analyses offers an efficient setup for assessing cardiac toxicity and, hence, can facilitate drug development [291].

A simple and low-cost zebrafish heart failure model has been utilized for drug discovery. When treated with aristolochic acid (AA), zebrafish embryos can induce cardiac defects that resemble heart failure. A drug-screening assay using AA-treated zebrafish embryos identified that the mitogen-activated protein kinase inhibitor (MEK-I), C25 (a chalcone derivative), and A11 (a phenolic compound) have the potential to attenuate heart failure by different mechanisms [300]. The study underscores the importance of drug-induced animal heart failure models for screening molecules with specific therapeutic functions. Zebrafish have been proposed as an efficient model for testing drugs with potential electrophysiological effects on cardiomyocytes [234,301].

A transgenic zebrafish (*Tg*(*fli1: eGFP*)) that stably expresses eGFP within vascular endothelial cells has been used to develop a high-content-screening (HCS) assay that can screen and identify chemicals affecting cardiovascular functions at sublethal, nonteratogenic concentrations. Treated zebrafish embryos were subjected to automated image acquisition procedures and custom image analysis to measure parameters such as body length, circulation, heart rate, pericardial area, and inter-segmental vessel area. The assay thus provides a complete discovery platform to identify chemicals that target cardiovascular functions at nonteratogenic concentrations, with important advantages of increased sample sizes and a broad concentration–response design [302]. In the field of drug discovery, zebrafish have provided significant contributions. More recently, the specific activity of tolterodine toward muscarinic M3 receptors over M2 receptors in heart development has been identified [303]. Various pharmacological reagents have already been screened using the zebrafish system, and these screenings have yielded numerous compounds beneficial for the cardiovascular system. Very recently, a fully automated in vivo screening system (AISS) enabling the rapid evaluation of the biological responses of non-anesthetized zebrafish to molecular gradients has been developed [293]. This tool will offer substantial potential for cost-effective and time-efficient handling, transportation, and assessment of zebrafish, introducing a novel approach to automated drug screening.

## 4. Part III

### 4.1. Recent Advancements in Zebrafish Research Toolkits

In recent years, rapid advancements have been made in zebrafish research, particularly in cardiac biology. Developing innovative genetic tools, including CRISPR/Cas9 and other genome-editing technologies, has allowed for a more precise understanding of heart development, disease, and regeneration. The following sections highlight the key studies from various research areas to showcase the significant contributions that zebrafish models have made to the field.

#### 4.1.1. CRISPR/Cas9-Mediated Gene Editing for Cardiac Research

CRISPR/Cas9 technology has revolutionized the study of cardiac development and disease. One of the earliest and most significant applications in zebrafish was the conditional tissue-specific ablation of genes to study their roles in heart function. For instance, Angom et al. (2023) used CRISPR/Cas9 to knockout the *nrp1b* isoform, specifically, in cardiomyocytes conditionally [172]. This study showed that *nrp1b* plays crucial roles in cardiac remodeling and heart function, with *nrp1b*-deficient zebrafish displaying impaired heart regeneration and altered cardiac morphology. This finding underscores the potential of CRISPR/Cas9 for studying gene function in the heart in vivo. Similarly, Cui et al. (2023) employed CRISPR/Cas9 to generate homozygous *kcnq1* deletion in zebrafish, which models type 1 long QT syndrome (LQTS) [304]. The researchers further expressed human *Kv7.1/MinK* channels in *kcnq1-*deficient embryos, providing a valuable in vivo model for studying cardiac arrhythmias. These models have enabled a better understanding of ion channelopathies and how specific mutations affect cardiac rhythm.

#### 4.1.2. Lineage Tracing and Cellular Origins of Cardiac Repair

Lineage-tracing techniques have been pivotal in identifying the cellular origins of cardiac repair and regeneration in zebrafish. The zebrafish heart is known for its remarkable regenerative abilities, and recent advancements have provided more profound insights into this process. For example, Li et al. (2022) utilized dachsous-cadherin-related 2 (DCHS2), a downstream effector protein, to study its effects on cardiac hypertrophy [305]. Using both zebrafish and mouse models, the researchers demonstrated that the transgenic overexpression of DCHS2 led to pathological hypertrophy in zebrafish, which impaired cardiac regeneration, while the same effect in mice led to physiological hypertrophy and promoted cardiomyogenesis. This finding illustrates the utility of zebrafish in modeling the cellular mechanisms of hypertrophy and regeneration. A significant contribution to understanding heart regeneration comes from the study by Jopling et al. (2010), where they demonstrated that zebrafish cardiomyocytes can proliferate and regenerate following heart injury [306]. Their work uncovered important pathways, such as the activation of notch signaling, in promoting cardiac regeneration. This study, along with another by Li et al. (2022), highlights how lineage-tracing technologies in zebrafish can pinpoint the cellular sources of regeneration in response to injury [148]. In addition, Kemmler et al. (2021) integrated transgenic reporters with CRISPR/Cas9 genome engineering to enable functional assessments of the genes involved in congenital heart defects [307]. Their study identified critical genes that govern heart field migration and the differentiation of myocardial progenitors, thus improving our insight into the molecular mechanisms behind inherited heart defects. Another related study by Targoff et al. (2013) used in vivo lineage tracing to study the migration of epicardial progenitors, which are crucial for heart repair after injury [72].

#### 4.1.3. High-Throughput Lineage Tracing with GESTALT

A landmark advancement in high-throughput lineage tracing was the introduction of the GESTALT (genome editing of synthetic target arrays for lineage tracing) system by McKenna et al. (2016) [308]. This powerful system allows the sampling of thousands of genes from different cellular populations within zebrafish, enabling the creation of comprehensive cellular lineage maps in various organs, including the heart. The integration of GESTALT has enabled large-scale analyses of cellular behaviors during heart development and disease progression. The system can trace specific high-resolution cellular lineages and identify subtle changes in cardiac morphogenesis and regeneration, which may be difficult to detect in more traditional models. Raj et al. (2025) further expanded on this by optimizing scGESTALT v2, which improves single-cell resolution in lineage tracing and provides higher-quality data during regeneration [309]. These advancements have provided more robust tools for studying cellular heterogeneity in the heart and other organs.

#### 4.1.4. The LINNAEUS System for Simultaneous Lineage Tracing and Transcriptome Profiling

The LINNAEUS system is a major breakthrough in simultaneously performing lineage tracing and transcriptome profiling [310]. CRISPR/Cas9-based nuclease-activated editing allows for real-time lineage tracing in zebrafish larvae and the simultaneous analysis of gene expression in tracked cells. This technology has been crucial in identifying dynamic cellular transitions during cardiac development. Spanjaard et al. (2025) have further optimized this system, improving the precision and scalability of lineage tracing in zebrafish [311]. The optimization of CRISPR/Cas9 systems for better temporal resolution has enabled researchers to capture dynamic developmental processes in the heart and other organs. By mapping developmental lineage trees, they were able to track the origins of novel cellular types involved in cardiac regeneration, providing new insights into the cellular plasticity that drives heart repair [311].

#### 4.1.5. TEMPO for Real-Time Imaging

Although most cardiac studies in zebrafish focus on heart-specific tools, advancements in neural imaging also offer insights into the broader biological processes that might influence cardiac development. The TEMPO (temporal encoding and manipulation in a predefined order) system, introduced by Espinosa-Medina et al. (2023), enables the real-time imaging of zebrafish neurogenesis using differential labeling and the manipulation of consecutive cellular generations [312,313]. This tool has been used to continuously observe fluorescent transitions in live zebrafish embryos, revealing how neurogenesis influences the development of other organ systems, including the heart. The ability to track cellular changes over time in a living organism presents exciting new opportunities to study developmental processes that span multiple organ systems.

#### 4.1.6. FRaeppli Toolbox for Multi-Spectral Imaging and Cellular Tracking

The FRaeppli toolbox, developed by Caviglia et al. (2022), represents a powerful advancement in zebrafish multi-spectral imaging and cellular tracking [314]. FRaeppli allows for precisely labeling different cardiac cellular populations using distinct fluorescent markers, enabling the detailed tracking of cellular interactions and tissue architecture at single-cell resolution. This technology is instrumental in studying the complexities of cardiac cellular interactions, such as those observed during heart regeneration and disease. In their work, Caviglia et al. (2022) demonstrated that FRaeppli could be used to analyze cardiac morphogenesis and tissue reorganization following injury, thus offering deeper insights into the cellular events that underlie heart development and repair [314].

#### 4.1.7. Cre/Lox System for Spatial and Temporal Control in Cardiac Studies

The Cre/lox system has been an indispensable tool in zebrafish models, allowing for the spatial and temporal control of gene expressions. Liu et al. (2022) developed a novel genetic mosaic toolkit, marking a significant advancement in zebrafish genetics. Their system allows for the precise manipulation of the genes involved in cardiac development and disease, such as those regulating cardiac conduction and myocardial differentiation [315]. The development of this toolkit enhances our ability to study gene functions in specific tissues and at precise developmental stages, providing insights into how cardiac diseases manifest and progress. Moreover, CRISPR/Cas9-mediated targeted knock-ins have been used to generate CreERT2 drivers regulated by endogenous gene elements, as shown in studies by Almeida et al. (2021) [316] and Kesavan et al. (2018) [317]. These tools provide unprecedented control over gene expressions, facilitating the study of gene functions in cardiac disease models and their responses to injury.

Next-generation single-cell sequencing (scRNA-seq) and multiomic approaches are poised to add a new dimension to zebrafish cardiovascular research. The advent of next-generation sequencing technologies, especially in the context of single-cell RNA sequencing, provides unprecedented resolution to examine the transcriptomic landscape of individual cells within the zebrafish heart [318,319,320]. This technology will allow us to identify rare and previously uncharacterized cellular types and trace the molecular events during disease progression and heart regeneration or development. Single-cell analysis can uncover the heterogeneity of cells within the heart, revealing how different cellular populations interact and contribute to the pathogenesis of cardiovascular diseases. These insights could identify novel molecular mechanisms and targets for therapeutic intervention. Moreover, combining single-cell sequencing with multiomic approaches, such as proteomics, metabolomics, and epigenomics, offers a robust framework to understand the full spectrum of molecular changes underlying cardiovascular disease [321]. The ability to perform spatial transcriptomics on zebrafish heart tissue is another exciting prospect that could significantly enhance our understanding of tissue-specific molecular processes. Spatial transcriptomics allows for mapping gene expression patterns directly onto tissue sections, which will be invaluable for studying the complex tissue architecture of the heart during disease or regeneration [322]. This technology could enable us to understand how specific cellular types in distinct heart regions contribute to disease and regeneration.

Integrating genomic technologies with high-throughput-screening platforms represents one of the most promising directions for the future of zebrafish cardiovascular research. This methodology could substantially speed up the discovery of novel therapeutics for cardiovascular diseases that presently lack for effective treatments. Furthermore, combining multiomic data with machine learning and artificial intelligence (AI) could facilitate more accurate disease advancement and treatment response predictions. AI algorithms can integrate vast amounts of data from different omic layers to identify complex patterns, which could help to predict disease outcomes and identify the most promising therapeutic strategies for individual patients.

## 5. Conclusions

The zebrafish model is an important animal model for cardiac development, cardiac functions, and human heart diseases. In the past two decades, numerous zebrafish genetic screens have generated cardiac mutants, affecting several processes during early heart growth and chamber development. Combining this cardiac mutant with high-resolution and advanced (3D and 4D) imaging tools will make it easier to follow the details of cardiac development. Also, genome-editing tools in zebrafish biology are developing paramountcy in heart development and function. The progress in gene-editing and -targeting technologies has provided a scope for functional studies of genes and variations implicated in CVDs in zebrafish. The fish model is being utilized to investigate adult cardiac disorders, as it provides the advantage of studying adult heterozygous fish to gain insights into the progression of cardiac disease and its pathogenesis. Further, zebrafish have made significant contributions to the field of drug discovery, and it is expected that these efforts will soon generate a library of compounds with cardiovascular-disease-monitoring properties. This is strongly supported by pharmacological screens that have yielded compounds beneficial for managing or treating cardiovascular disorders. Altogether, zebrafish cardiovascular research has provided evidence to fill the knowledge gap in understanding human and other mammalian cardiac development and disorders. Although resources, including reverse and forward genetics, ZFNs, TALENs, TILLING, and CRISPR, have contributed significantly to zebrafish cardiovascular research, the field is advancing toward more sophisticated technologies that enable higher precision, complexity, and integration. As the tools for genetic manipulation evolve, researchers must continue to familiarize themselves with and embrace these new technologies to discover more profound understandings of cardiovascular diseases and develop more effective therapies. The future of zebrafish cardiovascular research lies in integrating next-generation genome-editing technologies, multi-omics, and AI, delivering more inclusive knowledge about cardiovascular biology and human diseases, eventually leading to innovative treatment strategies.

## Figures and Tables

**Figure 1 cells-14-00531-f001:**
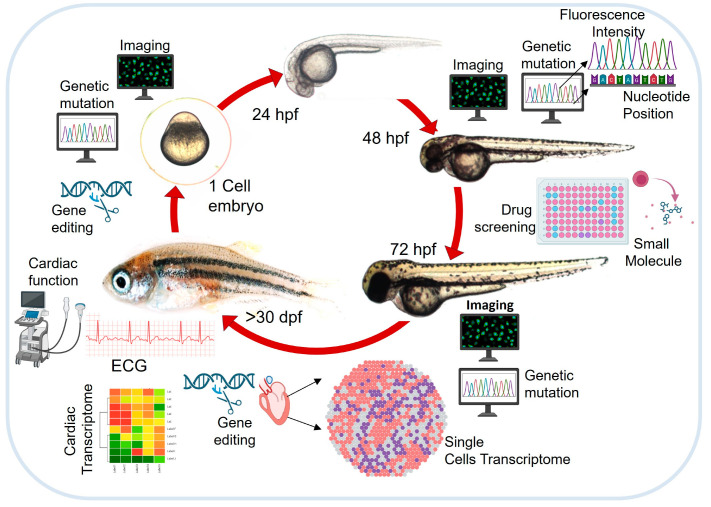
Schematics showing zebrafish developmental stages and application as a multifaceted model for cardiovascular research. The image was created using BioRender (https://www.biorender.com/). hpf: hours post fertilization; dpf: days post fertilization.

**Figure 2 cells-14-00531-f002:**
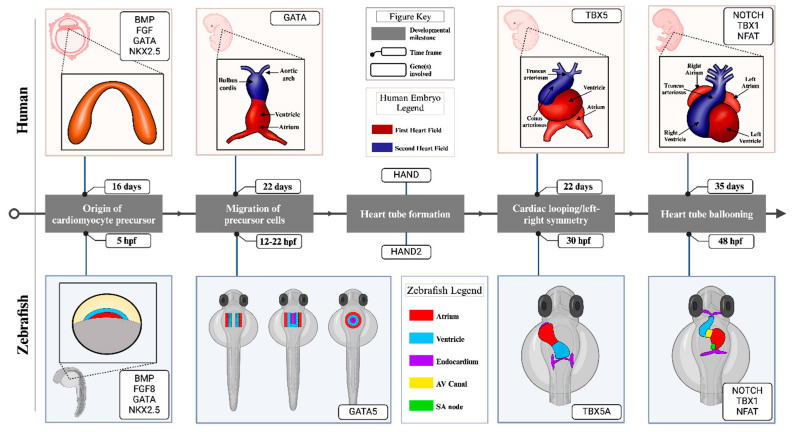
Schematics showing the comparative stages of heart development in humans and zebrafish.

**Figure 3 cells-14-00531-f003:**
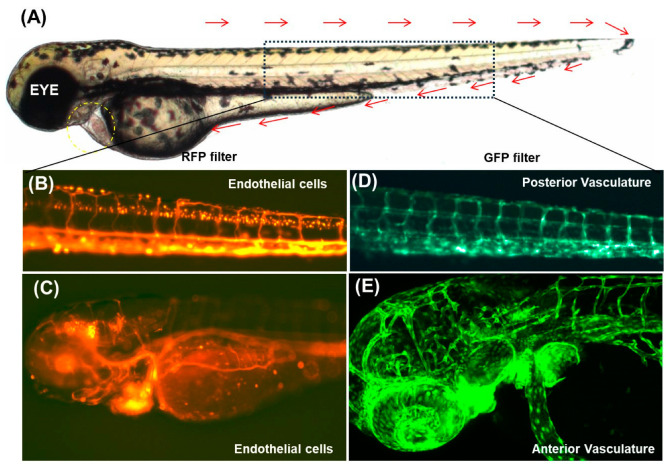
Transgenic zebrafish larva. (**A**) Bright-field image of a 3-day-old *Tg(fli: eGFP; Gata1: DsRed)* zebrafish larva. (**B**) The trunk region of the zebrafish larva shows red fluorescence protein expression in the blood cells. (**C**) Anterior region showing blood cells under an RFP filter. (**D**) Trunk region expressing green fluorescence protein in the ISVs. (**E**) Anterior region expressing green fluorescence protein in the endothelial cells. The images presented were generated in the authors’ laboratory, using a Zeiss Axio observer 2.1 microscope and a bright-light filter (**A**), an RFP filter (568 nm) (**B**,**C**), and a GFP filter (488 nm) (**D**) and a confocal microscope (LSM 880) and 488 nm wavelength lasers (**E**).

**Figure 4 cells-14-00531-f004:**
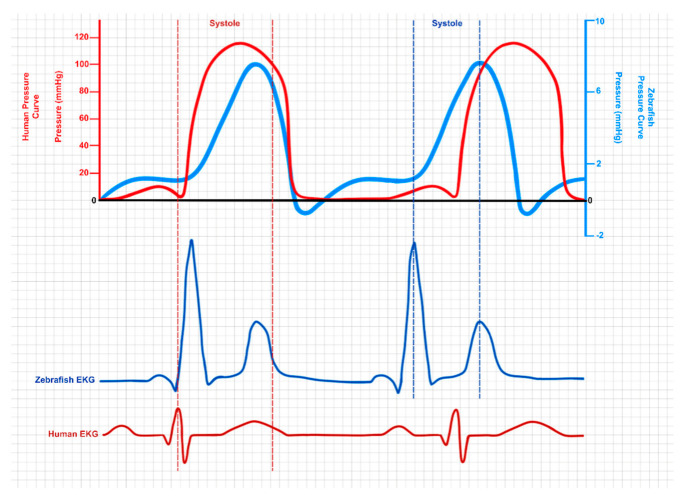
Comparison of zebrafish and human cardiovascular dynamics, demonstrating analogous systolic and diastolic phases with pulsatile blood flow and similar ventricular depolarization and repolarization patterns in their ECGs, highlighting the key shared hemodynamic and electrophysiological features.

**Table 1 cells-14-00531-t001:** Overlap in heart development between humans and zebrafish.

1	**Specification of Cardiac Progenitor Cells**			
	Human	Zebrafish	Signaling Pathways	References
	Specification of mesodermal cells to become cardiac progenitors and migration to form the cardiac crescent (early heart field)	The lateral plate mesoderm gives rise to bilateral populations of cardiac progenitor cells that migrate toward the midline.	BMP, FGF, and Wnt signaling	[31,36]
2	Migration and Formation of the Heart Tube			
	Migration of the progenitor cells to the midline and formation of a linear heart tube	The bilateral heart fields migrate and merge at the midline, creating a single-layered primitive heart tube.	Wnt, VEGF, and nodal signaling	[37,38]
3	Cardiac Looping			
	The linear heart tube elongates and bends to the right (dextral looping), establishing the left–right asymmetry critical for chamber formation.	The primitive heart tube loops in a similar manner but occurs on a shorter timescale.	Pitx2 and nodal signaling	[39,40]
4	Chamber Formation and Septation			
	The heart tube differentiates into atrial and ventricular chambers. Endocardial cushions contribute to septation, forming the four-chambered heart.	Zebrafish form two chambers (atrium and ventricle) without septation. The atrial and ventricular compartments are morphologically distinct.	Tbx, Nkx2.5, and hand transcription factors	[41,42,43]
5	Trabeculation and Valvular Formation			
	The ventricles develop trabeculae (muscular ridges), enhancing contractility. Valves develop from endocardial cushions.	Trabeculation occurs in the ventricle, and valves develop at the atrioventricular junction.	Notch, Bmp, and ErbB pathways	[44,45]
6	Maturation and Growth of the Heart			
	The heart continues to grow through cardiomyocyte proliferation and maturation, and coronary vessels form to supply the heart muscle.	Growth primarily occurs through cardiomyocyte proliferation, and coronary vasculature develops later.	IGF and VEGF	[26,43,46]

**Table 3 cells-14-00531-t003:** Comprehensive overview of research tools and applications of zebrafish in cardiovascular research.

Experimental Technique	Approach	Application	References
* Genetic manipulation techniques *	CRISPR/Cas9	High-throughput screens in cardiomyocytes, multigene targeting, heat-shock-inducible CRISPR/Cas9 for cardiomyocyte-specific knockouts	[171,172]
	Transgenic lines	Tg(myl7:GA): calcium dynamics and heart failure modeling, Casper (roy−/−, nacre−/−): cardiac injury and regeneration modeling, transposon-based transgenesis (Tol2 system): cardiac calcium dynamic mapping	[173,174,175]
	Zinc finger nucleases (ZFNs)	Targeted germline mutagenesis (heritable gene disruption (e.g., golden and no-tail genes), vascular development studies (e.g., gata2a mutation))	[176,177]
	Transcription activator-like effector nucleases (TALENs)	High-efficiency gene knockout, expanded targeting capabilities (e.g., CpG-rich and start codon regions)	[178,179]
* Imaging techniques *	Confocal microscopy and high-speed imaging	Real-time vascular dynamics and endothelial function imaging	[180]
	Ultrasound bio-microscopy	Cardiac structure imaging and blood flow measurements	[181]
	Fluorescence imaging	Glycol methacrylate (GMA) embedding as a practical approach for studying deeper structures (e.g., vasculatures)	[182]
	Light-sheet fluorescence microscopy	High-resolution, real-time imaging of zebrafish cardiac development, blood flow, and biomechanical forces (e.g., wall shear stresses)	[183]
* Pharmacological utility *	Small-molecule screening/small-compound screening	Drug discovery and vascular disease modeling	[184,185]
	Retro-orbital microinjection, immersion-based drug delivery, and pseudodynamic 3D imaging of cardiac functions	Drug response evaluation (of cardiac functions to model cardiomyopathy), cardiotoxicity testing, and cardiac parameter quantification	[186]
	Drug exposure assays and high-throughput screening	Drug-induced toxicity screening and heart rate and rhythm analyses	[187]
	In vivo zebrafish cardiovascular assays	Drug effect evaluation, high-throughput screening, and mechanistic insights	[188]

## Data Availability

No new data were generated in this study.

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
