# Peer review of "Zebrafish as a Versatile Model for Cardiovascular Research: Peering into the Heart of the Matter"

_cells, 2025, doi:10.3390/cells14070531_

Round 1
Reviewer 1 Report (Previous Reviewer 2)
Comments and Suggestions for Authors
The review article "Zebrafish as a Versatile Model for Cardiovascular Research: Peering into the Heart of the Matter Ramcharan" by Singh Angom * , Meghna Singh , Huzaifa Muhammad is a sound and extensive summary of the research of heart development and heart disease in the past 15-20 years. The authors addressed some of the points this reviewer criticized previously by adding a new section with recent advances. But also there, it is just a summary without any new knowledge. Just adding a section is not enough to qualify this as an innovative and novel review article.
Therefore, the main point is still standing, almost all the information can be found in numerous other reviews just like this one. This reviewer is still of the opinion that this is more appropriate for a textbook than it is as a research/review article. Specifically, the sections about reverse and forward genetics and zincfinger, TALENs, TILLING and even CRISPR are too exhaustive. All these topics have been extensively reviewed in the last 10 years and no new knowledge can be gained from this review article. I’m also missing your personal perspective, where do you see the field going with all these new technologies? Which doors will be opened for new research approaches? What do you think might be possible in future and wasn’t possible until now? Does this help with curing heart disease by generating more accurate models or gain better understanding of genetics and mechanisms of disease?
Author Response
Comments: The review article "Zebrafish as a Versatile Model for Cardiovascular Research: Peering into the Heart of the Matter Ramcharan" by Singh Angom * , Meghna Singh , Huzaifa Muhammad is a sound and extensive summary of the research of heart development and heart disease in the past 15-20 years. The authors addressed some of the points this reviewer criticized previously by adding a new section with recent advances. But also there, it is just a summary without any new knowledge. Just adding a section is not enough to qualify this as an innovative and novel review article.
Response: We appreciate the reviewer’s feedback and suggestions. The newly added section was intended to provide a comprehensive overview of recent advances while ensuring a logical flow with the rest of the review. However, we recognize that simply summarizing existing knowledge may not sufficiently demonstrate novelty. To address this, we have further expanded our discussion by critically analyzing these recent advances, identifying gaps in current knowledge, and proposing new perspectives or future research directions. We believe this addition strengthens the manuscript's contribution to the field. If the reviewer has specific aspects that they feel require deeper exploration or a different approach, we would be happy to incorporate additional insights accordingly. Please see the changes highlighted below. These changes has been included in the text from Line 1031-1166, and 1186-1195.
Comment # 2: The main point is still standing, almost all the information can be found in numerous other reviews just like this one. This reviewer is still of the opinion that this is more appropriate for a textbook than it is as a research/review article. Specifically, the sections about reverse and forward genetics and zincfinger, TALENs, TILLING and even CRISPR are too exhaustive. All these topics have been extensively reviewed in the last 10 years and no new knowledge can be gained from this review article. I’m also missing your personal perspective, where do you see the field going with all these new technologies? Which doors will be opened for new research approaches? What do you think might be possible in future and wasn’t possible until now? Does this help with curing heart disease by generating more accurate models or gain better understanding of genetics and mechanisms of disease?
Response: We appreciate the reviewer’s feedback and understand the concern that our review may be perceived as too descriptive rather than providing a novel perspective. In our previous revision, we tried to provide comprehensive and updated information of genetic tools in zebrafish cardiovascular research, but we admit that certain sections may be overly exhaustive. To address these concerns, we have reorganized and updated our review as shown below. We hope these revisions significantly improve the manuscript’s contribution to the field and will address the reviewer’s concerns.
We welcome additional guidance on specific aspects the reviewer believes should be further emphasized.
Please see the detailed response attached.

Reviewer 2 Report (Previous Reviewer 1)
Comments and Suggestions for Authors
The authors made extensive corrections to the previous manuscript version. I have a few additional comments to improve the manuscript:
Line 157: the last sentence does not link well to the previous discussion of FHF and SHF. Please add an explanation how these parts of the heart relate to the discussion in the same paragraph.
Figure 2 legend - There is still too much description of the graphical mechanics employed, which should be removed. Instead, a high-level description of what is shown in the respective parts of the figure should be provided.
Line 234: The phrase "distinctly studying" seems a little cumbersome. Maybe "isolating" or "identifying" could be better.
Line 297: "Differentiation and morphogenesis" of what?
Line 356: "Inhibition" is incorrect here. Please see the abstract excerpt here: "The dual specificity phosphatase 6 (Dusp6) functions as a feedback regulator of fibroblast growth factor (FGF) signaling to limit the activity of extracellular signal regulated kinase (ERK) 1 and 2. We have identified a small molecule inhibitor of Dusp6, (E)-2-benzylidene-3-(cyclohexylamino)-2,3-dihydro-1H-inden-1-one (BCI), using a transgenic zebrafish chemical screen. BCI treatment blocked Dusp6 activity and enhanced FGF target gene expression in zebrafish embryos."
Line 705: Please use correct formatting of transgenic line names. Compound line names should be separated in some way.
Line 1127: "zebra fish" should be "zebrafish".
Author Response
Reviewqer #2
The authors made extensive corrections to the previous manuscript version. I have a few additional comments to improve the manuscript:
Response: Thank You for all the valuable feedback.
Line 157: the last sentence does not link well to the previous discussion of FHF and SHF. Please add an explanation how these parts of the heart relate to the discussion in the same paragraph.
Response: Thank You for this correction. We have now added the complete sentence. Please see the changes in Line 157-160.
Figure 2 legend - There is still too much description of the graphical mechanics employed, which should be removed. Instead, a high-level description of what is shown in the respective parts of the figure should be provided.
Response: Thank You for this suggestion. We have removed extra information as shown in the updated manuscript. We hope this changes addressed the reviewres concerns.
Line 234: The phrase "distinctly studying" seems a little cumbersome. Maybe "isolating" or "identifying" could be better.
Response: Thank you for this suggestion. We have made this changes in Line 235.
Line 297: "Differentiation and morphogenesis" of what?
Response: We have corrected the missing information: “Cardiac Progenetor cells. Please see the changes in line 298-299.
Line 356: "Inhibition" is incorrect here. Please see the abstract excerpt here: "The dual specificity phosphatase 6 (Dusp6) functions as a feedback regulator of fibroblast growth factor (FGF) signaling to limit the activity of extracellular signal regulated kinase (ERK) 1 and 2. We have identified a small molecule inhibitor of Dusp6, (E)-2-benzylidene-3-(cyclohexylamino)-2,3-dihydro-1H-inden-1-one (BCI), using a transgenic zebrafish chemical screen. BCI treatment blocked Dusp6 activity and enhanced FGF target gene expression in zebrafish embryos."
Response: We appreciate the reviewer for this suggestion. We have corrected this as suggested. Pleas esee this changes in line 357.
Line 705: Please use correct formatting of transgenic line names. Compound line names should be separated in some way.
Response: We have corrected the name. Tg(fli:eGFP;Gata1:DsRed)
Line 1127: "zebra fish" should be "zebrafish".
Response: Thank You. We have corrected this in new Line 1132.

This manuscript is a resubmission of an earlier submission. The following is a list of the peer review reports and author responses from that submission.
Round 1
Reviewer 1 Report
Comments and Suggestions for Authors
The review is an interesting and comprehensive overview of the field from a particular standpoint of the authors. Although there are some quality issues in the text, there are many good ideas and conclusions that improve the reader's understanding of the field. It would be great to have a specific concept advanced by the review that would distinguish it from other similar works, but this may not always be possible. The following are my comments in the order of the text, but without specific categories of importance:
1. Lines 43-44: Some inconsistency in gene names and abbreviations. This is an issue throughout the manuscript so please review the nomenclature to mention the genes correctly.
2. Line 50: "to be mapped" should be "to map".
3. Lines 55-56: The phrase "humans and other vertebrate species such as rats, mice, pig/ swine, dog/canine, and zebrafish" can be replaced by "vertebrates".
4. Line 57: "recreate" should be "model".
5. Line 59: Check the format for citations.
6. Line 64: The first diagram is not informative.
7. Line 67: What is the strain "ASWT"?
8. Line 80: How does the phrase " more analogous features" relate to homology?
9. Lines 82-83: The phrase "presence of an atrium as a protective pericardial sac in the thoracic cavity is common" is not clear in terms of the meaning and the animal model.
10. Line 95: "made in" should be "to".
11. Line 117: The expression "20-something" does not make sense.
12. Table 1 - A list of pathways or molecules does not make a mechanism.
13. Line 128: the title does not match the content of the paragraph.
14. Lines 157-158: the sentence does not make sense without further explanation. Please correct this.
15. Figure 2:
- No overall context of heart in human embryo is provided.
- The coloring scheme in humans is not clear.
- Please define the colors and species for the genes indicated in the figure
- The correspondence between stages in two species is unclear.
- The legend is not appropriate since how the figure was made is not important in a scientific paper.
16. Lines 164-167: stage names are incorrect.
17. Page 7: Gene naming is not always consistent here.
18. Line 207: What "is an advantage" here?
19. Page 8: gene naming is not followed correctly.
20: Page 8: The sentence "This is well observed in the case when mice mutants lacking Nkx2-5 do not produce a typical fly heartless (tinman) phenotype" is strange. How are mice supposed to produce a fly phenotype?
21. Lines 317-318: The phrase "Feedback inhibition of fgf signaling by a small molecule antagonist of dual specificity phosphatase (Dusp6 ) " is contradictory. Inhibition of an inhibitor = activation, but this sentence does not indicate this.
22. Line 335: "BMPsare" should be "BMPs are".
23. Lines 354-355; The phrase "It is in differentiating cardiomyocytes." is problematic since it does not serve clear purpose.
24. Line 367: Variant RNA definition is not worded correctly.
For further issues found in this manuscript, please check the attached file.

Author Response
Reviewer #1
The review is an interesting and comprehensive overview of the field from a particular standpoint of the authors. Although there are some quality issues in the text, there are many good ideas and conclusions that improve the reader's understanding of the field. It would be great to have a specific concept advanced by the review that would distinguish it from other similar works, but this may not always be possible. The following are my comments in the order of the text, but without specific categories of importance:
Response: Thank you for your thoughtful and constructive feedback on our review. We appreciate your acknowledgment of the comprehensiveness of our work and are pleased that you found it to contain valuable ideas and conclusions that enhance the understanding of the field. We understand the importance of providing a distinct concept that sets this review apart from similar works. While the primary aim of our review is to summarize recent advances, we carefully considered how we can further highlight unique perspectives or novel concepts to enhance their impact. We also acknowledge the quality issues mentioned in the text and have addressed these in our revisions to ensure clarity and precision. We appreciate the detailed comments provided in the order of the text, and we will carefully address each point to improve the manuscript further. The authors thank the reviewer for the valuable feedback, which will help us refine and strengthen our review.
- Lines 43-44: Some inconsistency in gene names and abbreviations. This is an issue throughout the manuscript so please review the nomenclature to mention the genes correctly.
Response: Thank You. We have corrected these mistakes in the manuscript.
- Line 50: "to be mapped" should be "to map".
Response: We have corrected this.
- Lines 55-56: The phrase "humans and other vertebrate species such as rats, mice, pig/ swine, dog/canine, and zebrafish" can be replaced by "vertebrates".
Response: Thank you. We have changed as per suggestion.
- Line 57: "recreate" should be "model".
Response: We have now corrected it.
- Line 59: Check the format for citations.
Response: We have now corrected it.
- Line 64: The first diagram is not informative.
Response: We have now corrected it.
- Line 67: What is the strain "ASWT"?
Response: We have now provided the information in the manuscript it.
- Line 80: How does the phrase " more analogous features" relate to homology?
Response: We changed it with a comparable.
- Lines 82-83: The phrase "presence of an atrium as a protective pericardial sac in the thoracic cavity is common" is not clear in terms of the meaning and the animal model.
Response: We have now corrected our sentence as “The mediolateral location of the ventricle, the presence of an atrium, and a protective pericardial sac in the thoracic cavity are common”. We hope this change will address the reviewer’s concern..
- Line 95: "made in" should be "to".
Response: Thank You. We have replaced it.
- Line 117: The expression "20-something" does not make sense.
Response: We have corrected this.
- Table 1 - A list of pathways or molecules does not make a mechanism.
Response: We have changed this.
- Line 128: the title does not match the content of the paragraph.
Response: We have changed the title as per suggestion.
- Lines 157-158: the sentence does not make sense without further explanation. Please correct this.
Response: We have removed this sentence to avoid any distraction.
- Figure 2:
- No overall context of heart in human embryo is provided.
- The coloring scheme in humans is not clear.
- Please define the colors and species for the genes indicated in the figure
- The correspondence between stages in two species is unclear.
- The legend is not appropriate since how the figure was made is not important in a scientific paper.
Response: We have addressed all the changes with appropriate information in figure 2.
a.In Figure 1, Various information, including tools and techniques, has been incorporated at each stage. This demonstrated the utility of zebrafish for CVD research.
- In Figure 2, Corresponding human embryo illustrations have been added to each stage in the human heart development section to provide overall anatomical context and clarify the progression of cardiac development.
- In Figure 2, genes have been clearly placed on opposite sides of the timeline (on respective sides for humans and zebrafish) next to the stage of their developmental contribution to clarify their species correspondence.
- A timeline has been implemented into the designed to portray developmental progression and explicitly link the developmental stages of humans and zebrafish, with corresponding stages aligned vertically to emphasize their correlation. The time frame callout symbols for each species emerge from a common origin on the timeline, visually reinforcing their developmental parallels and demonstrating their correspondence, despite differing time scales (days for humans and hpf for zebrafish), within the overall context of heart development
- Lines 164-167: stage names are incorrect.
Response: We have made the suitable changes.
- Page 7: Gene naming is not always consistent here.
Response: We have made the changes to correct the gene names.
- Line 207: What "is an advantage" here?
Response: We have modified the sentence as “Unlike chicks, zebrafish embryos can develop a heart even in the absence of endoderm, which provides an advantage for distinctly studying the cardiogenic role of Gata5 without complications arising from its additional roles in the endoderm”. We hope that this sentence will convey the message clearly.
- Page 8: gene naming is not followed correctly.
Response: We have corrected the gene names.
20: Page 8: The sentence "This is well observed in the case when mice mutants lacking Nkx2-5 do not produce a typical fly heartless (tinman) phenotype" is strange. How are mice supposed to produce a fly phenotype?
Response: Thank You for this suggestion. To convey the correct message, we have now changed the sentence with this version: “This is evident in mouse mutants lacking Nkx2-5, which do not exhibit the typical fly 'heartless' (tinman) phenotype but instead highlight a role for Nkx2-5 in differentiation and morphogenesis”.
- Lines 317-318: The phrase "Feedback inhibition of fgf signaling by a small molecule antagonist of dual specificity phosphatase (Dusp6 ) " is contradictory. Inhibition of an inhibitor = activation, but this sentence does not indicate this.
Response: Thank You. We have now replaced this sentence with “Inhibition of fgf signaling by a small molecule antagonist of dual specificity phosphatase (Dusp6) increases the heart field size resulting in an enlarged heart suggesting that Dusp6 functions as an attenuator of FGF signaling in the cardiac field to regulate heart organ size”
- Line 335: "BMPsare" should be "BMPs are".
Response: We have corrected it.
- Lines 354-355; The phrase "It is in differentiating cardiomyocytes." is problematic since it does not serve clear purpose.
Response: Thank You. We change it to “In differentiating cardiomyocytes, BMP signaling regulates myocardial differentiation by inducing the expression of tbx20 and tbx2b at the mid-somite stages while simultaneously undergoing inhibition via smad6.
- Line 367: Variant RNA definition is not worded correctly.
Response: We wrote When injected into zebrafish embryos, the ALK2 variant L343P RNA led to improper atrioventricular canal formation.
Reviewer 2 Report
Comments and Suggestions for Authors
The review article "Zebrafish as a Versatile Model for Cardiovascular Research: Peering into the Heart of the Matter Ramcharan" by Singh Angom * , Meghna Singh , Huzaifa Muhammad is a sound and extensive summary of the research of heart development and heart disease in the past 15-20 years. The article is well written and well structured. However, there is nothing new in there, as can be seen by the literature cited, of which 90% is over 5 years old. Most of the figures have been published in this or an adapted version before. The methods described are not cutting-edge and several are missing like knock-in via CRSIPR/Cas9 for transgenesis. Tissue specific knock-out, over-expression and lineage tracing techniques that are emerging are not mentioned at all, e.g. TeON/OFF or the Cre/lox system. Overall, I am missing new information or a unique perspective that the authors may offer for new insights that are of interest to the field.
Comments on the Quality of English LanguageLanguange is appropriate.
Author Response
Reviewer #2
The review article "Zebrafish as a Versatile Model for Cardiovascular Research: Peering into the Heart of the Matter Ramcharan" by Singh Angom * , Meghna Singh , Huzaifa Muhammad is a sound and extensive summary of the research of heart development and heart disease in the past 15-20 years. The article is well written and well structured.
Response: "Thank you for your kind and encouraging feedback on our review article. We are delighted that you found it to be a comprehensive and well-structured summary of the advancements in heart development and cardiovascular research. Your positive remarks are greatly appreciated and motivate us to continue contributing to the field.
- However, there is nothing new in there, as can be seen by the literature cited, of which 90% is over 5 years old.
Response: We thank the reviewer for their comment. To address this concern, we have updated our manuscript with recent literature wherever possible and relevant. We believe these updates have significantly improved the quality and relevance of our review.
- Most of the figures have been published in this or an adapted version before.
Response: Thank you for bringing this to our attention. We have extensively revised Figures 1 and 2 to better align with this review's core message and provide a fresh perspective. We hope these revisions adequately address your concern. Please find the detailed changes below.
a.In Figure 1, Various information, including tools and techniques, has been incorporated at each stage. This demonstrated the utility of zebrafish for CVD research.
- In Figure 2, Corresponding human embryo illustrations have been added to each stage in the human heart development section to provide overall anatomical context and clarify the progression of cardiac development.
- In Figure 2, genes have been clearly placed on opposite sides of the timeline (on respective sides for humans and zebrafish) next to the stage of their developmental contribution to clarify their species correspondence.
- In Figure 2, A timeline has been implemented into the designed to portray developmental progression and explicitly link the developmental stages of humans and zebrafish, with corresponding stages aligned vertically to emphasize their correlation. The time frame callout symbols for each species emerge from a common origin on the timeline, visually reinforcing their developmental parallels and demonstrating their correspondence, despite differing time scales (days for humans and hpf for zebrafish), within the overall context of heart development
- The methods described are not cutting-edge and several are missing like knock-in via CRSIPR/Cas9 for transgenesis. Tissue specific knock-out, over-expression and lineage tracing techniques that are emerging are not mentioned at all, e.g. TeON/OFF or the Cre/lox system.
Response: We appreciate the reviewer’s comment. We have now updated the manuscript and provided a separate section describing:” recent advancements in the field”. These includes some of the most recent techniques in the zebrafish cardiovascular research as suggested by the reviewer.
- Overall, I am missing new information or a unique perspective that the authors may offer for new insights that are of interest to the field.
Response. Thank you for your valuable feedback. We have revised the manuscript to include new data and updated the figures to provide additional insights. We believe these changes address your concerns and enhance the originality and relevance of our work, offering a fresh perspective that will be of interest to the field.
Reviewer 3 Report
Comments and Suggestions for Authors
Dear Authors,
My comments are in yellow and marked in the lines.
Kind regards, reviewer RK

Author Response
Reviewer #3
Dear Authors,
My comments are in yellow and marked in the lines.
Kind regards, reviewer RK
Response: "Thank you for your kind and encouraging feedback on our review article. Your positive remarks are greatly appreciated and motivate us to continue contributing to the field.
Response 1. We have updated the author's affiliations.
Response 2. We have removed additional references from line number 26.
Response 3. We have expanded the CVD to cardiovascular disease
Response 4. We have updated the gene name
Response. We have included the legend for Figure 1.
Response 5. We corrected the spelling for reference in Table 1.
Response 6. We have updated the version information for Figure 2.
Response 7. We have changed the sentence from italicized to regular text.
Response 8. The reference has been corrected
Response 9. We appreciate the reviewer's comment. We have included information in the figure legend to convey that these images were generated by the authors.
We have addressed all the comments as suggested by the reviewer and believe that the manuscript has significantly improved. We hope that our response adequately addresses the reviewer’s concerns
Round 2
Reviewer 3 Report
Comments and Suggestions for Authors
Dear Authors,
Here a few comments:
- 2 authors are listed out of the blue, in line 3 and 4?
- Line 672 till 691 seems not to have any references? Or is this completely own research?
- Line 709, the sentence stops with using...I assume there is something missing?
- Line 752 and 758 PhOTO is assume this should be photo?
- Lines (1136, 1138, 1143, and 1149) are programs, which are not appropriate listed as versions or either websites?
- Finally, in Table 2, 31126903 is not correct reference, which means your reference list needs to be adapted in the numbering for the rest!
Regards, reviewer